# Sea ice cover in the Copernicus Arctic Regional Reanalysis

Yurii Batrak[1], Bin Cheng[2], and Viivi Kallio-Myers[2]

[1]Development Centre for Weather Forecasting, Norwegian Meteorological Institute, Oslo, Norway
[2]Finnish Meteorological Institute, Helsinki, Finland

**Correspondence:** Yurii Batrak (yurii.batrak@met.no)

**Abstract.** The Copernicus Arctic Regional Reanalysis (CARRA) is a novel regional high-resolution atmospheric reanalysis product that covers a considerable part of the European Arctic including substantial amounts of ice-covered areas. Sea ice in CARRA is modelled by means of a one-dimensional thermodynamic sea ice parameterisation scheme, which also explicitly resolves the evolution of the snow layer over sea ice. In the present study we assess the representation of sea ice cover in CARRA and validate it against a wide set of satellite products and observations from ice mass balance buoys. We show that CARRA adequately represents general interannual trends towards thinner and warmer ice in the Arctic. Compared to ERA5, sea ice in CARRA shows a reduced warm bias in the ice surface temperature. The strongest improvement was observed for winter months over the Central Arctic, and the Greenland and Barents seas where a 4.91°C median ice surface temperature error of ERA5 is reduced to 1.88°C in CARRA on average. Over Baffin Bay, intercomparisons suggest the presence of a cold winter-time ice surface temperature bias in CARRA. No improvement over ERA5 was found in the ice surface albedo with spring-time errors in CARRA being up to 0.08 higher on average than those in ERA5 when computed against the CLARA-A2 satellite retrieval product. Summer-time ice surface albedos are comparable in CARRA and ERA5. Sea ice thickness and snow depth in CARRA adequately resolve the annual cycle of sea ice cover in the Arctic and bring added value compared to ERA5. However, limitations of CARRA indicate potential benefits of utilising more advanced approaches for representing sea ice cover in next generation reanalyses.

## 1 Introduction

Many scientific and engineering applications require, or can benefit from, information about the past and present states of the Earth's atmosphere provided by atmospheric reanalysis products (see, e.g., Frank et al., 2020; Chung et al., 2013; Serreze et al., 2003). These products utilise numerical weather prediction (NWP) systems constrained by a multitude of observational data to offer a pragmatic solution to the problem of obtaining a consistent multiyear gridded data set of atmospheric and surface variables. However, operational NWP systems applied in routine weather forecasting are under constant development, and, as a result, archives of operational weather forecasts comprise subsets produced with different versions of atmospheric models, each having its own biases and limitations, leading to inconsistent data sets. Therefore, to have a consistent gridded data set of

past atmospheric states, a series of objective analyses is repeated using the same version of an NWP system, which results in an atmospheric *reanalysis* data set (Bengtsson and Shukla, 1988).

Reanalysis systems are usually based on short range operational NWP systems (using an up-to-date model version at the time of the start of reanalysis production), which are kept unmodified. In the same way as the underlying NWP systems, atmospheric reanalysis data sets can be split into two categories: global and regional. Global reanalyses such as ERA5 (Hersbach et al., 2020), NCEP-DOE Reanalysis version 2 (Kanamitsu et al., 2002) or MERRA-2 (Gelaro et al., 2017) provide consistent gridded series of atmospheric and surface variables spanning multiple decades and covering the whole Earth. However, these reanalysis products usually have relatively coarse spatial resolution, ranging from hundreds of kilometres (in older products) to a few tens of kilometres in latest generation products (see, e.g., Fujiwara et al., 2017). Regional atmospheric reanalysis systems, unlike their global counterparts, are based on limited area NWP systems. Thus, they are less computationally expensive and allow higher spatial resolution and more advanced model formulations. Contemporary regional atmospheric reanalysis systems provide gridded data sets with spatial resolution close to 10 km and below (see, e.g., Kaiser-Weiss et al., 2019).

Ongoing climate change is leading to unprecedented modern-time warming of the Arctic, which is stronger than in any other region on Earth (Cohen et al., 2014; Serreze and Barry, 2011). Retreating sea ice and growing economic activity result in increasing scientific attention to the region, and demand on accurate and reliable atmospheric data sets. However, the Arctic is a challenging region to accurately model in NWP systems due to several factors. Firstly, the remote location of the Arctic limits the availability of in situ observations that can be used for constraining the models, although this lack of so-called conventional observations is partly compensated by higher availability of the satellite observations from polar orbiting satellites (Lawrence et al., 2019). Secondly, operational short range NWP systems that are used as the modelling component in atmospheric reanalysis systems are seldomly developed with the focus on resolving smaller-scale atmospheric processes typical for polar regions (Vihma et al., 2014). Additionally, even though accurate representation of surface processes in NWP systems is crucial in modelling interactions between the surface and the model atmosphere, numerical models often employ various simplifications to reduce computational costs, which can lead to increased modelling errors and biases. One example, more specific to the Arctic region, is the representation of sea ice cover. Sea ice, still abundant in the present-day Arctic, moderates the heat exchange between the ocean and the atmosphere, meaning that its accurate representation is vital to proper modelling of the surface energy balance, and, as a consequence, of near-surface atmospheric variables in the Arctic. Despite that fact, short-range NWP systems, and in turn, reanalysis systems that are based upon them, both global and regional, traditionally apply simplified one-dimensional parameterisation schemes for representing sea ice cover (Hines et al., 2015; Køltzow et al., 2019; Solomon et al., 2023). On the other hand, fully-coupled short-range NWP systems, which represent the atmosphere and the ocean three-dimensionally, as well as the dynamics and the evolution of sea ice cover, even though they are in the active development, to our knowledge are yet to be applied in contemporary atmospheric reanalysis systems (it must be noted, however, that fully coupled atmospheric reanalyses exist, for example, the Climate Forecast System Reanalysis (CFSR, Saha et al., 2010), albeit not based on a short range NWP system).

Contemporary atmospheric reanalysis systems utilise sea ice parameterisation schemes of various complexity ranging from those representing ice by means of computing the thermal balance of a thin ice layer (MERRA2) to thermodynamic sea ice

models, often with prescribed ice thickness (ASRv2, ERA5), or the snow layer omitted (ERA5). These schemes are developed with a focus on representing the surface energy balance of the ice layer since sea-ice specific variables such as ice thickness or ice salinity are of secondary interest in an atmospheric reanalysis. However, errors and biases found in the reanalysis products over the Arctic ocean (Graham et al., 2019; Wang et al., 2019) suggest potential benefits to implementing a more detailed representation of the evolution of sea ice in the current and next generation reanalysis systems.

In the present study we assess the performance of the Copernicus Arctic Regional Reanalysis (CARRA), a novel regional atmospheric reanalysis product for Greenland and the European Arctic based on the HAMRONIE-AROME NWP system (Bengtsson et al., 2017), in representing the evolution of sea ice cover. Additionally, we compare the representation of sea ice cover in CARRA, which employs a considerably more advanced sea ice parameterisation scheme, against the ERA5 reanalysis product used for obtaining lateral boundary conditions in the CARRA system. We focus on the following sea-ice specific variables: ice surface temperature, ice albedo, ice thickness and snow depth over sea ice. However, sea ice concentration, which is prescribed over the Arctic ocean in both CARRA and ERA5 from well-established satellite-based products is not validated in the present study. Near surface atmospheric variables over sea ice, such as two metre air temperature or ten metre wind speed, are not discussed in the present study mainly due to a limited number of available observations of these variables over sea ice within the area represented in CARRA. When performing comparisons, a wide set of remote sensing products is employed for assessing the performance of the reanalysis on the large scale. Additionally, observations reported from a set of ice mass balance buoys are used to complement comparisons against the remote sensing products.

The paper is organised as follows. Section 2 provides an overview of the studied atmospheric reanalysis products and underlying modelling systems, with special attention to the applied parameterisations of the sea ice cover. Section 3 describes the observational data sets utilised and the analysis methods applied in the present paper. Section 4 evaluates sea ice cover in CARRA and ERA5 by comparison against observational products. The final section provides a short summary of the obtained results and discusses their implications as well as opportunities for further improvements in representing sea ice cover in next generation reanalysis systems.

## 2 Representation of sea ice in CARRA and ERA5

### 2.1 CARRA

CARRA is a regional atmospheric reanalysis product which covers a sector of Arctic between 56°N and 86°N spanning from Baffin Bay in the west to the Kara Sea in the east. The data set covers the time period from September 1990 to the present (2023, at the moment of writing this manuscript) and analysis fields in CARRA are provided with a temporal resolution of 3 hours. In addition to objective analysis fields, the CARRA data set includes model forecasting data, which are provided with hourly temporal resolution for the lead times under 6 hours and with three-hourly resolution for lead times over 6 hours (lead times longer than 3 hours are available only for the forecasts initialised at 00 and 12 UTC). The CARRA system is based on the limited area NWP system HARMONIE-AROME (Bengtsson et al., 2017) and is forced by the ERA5 data on the model domain boundary. The reanalysis product is provided for two overlapping model domains: a larger western domain centred

on Greenland, and a smaller eastern domain covering the European Arctic (see Fig. 1). For both model domains a Lambert conformal conic grid with a horizontal resolution of 2.5 km is used.

Sea ice in the CARRA system is represented by a one-dimensional thermodynamic sea ice scheme (SICE, Batrak et al., 2018; Batrak and Müller, 2019) which resolves the processes of thermodynamic ice growth and melting. Snow cover on top of the sea ice is explicitly modelled by an adapted version of a multilayer parameterisation scheme originally developed for snow cover over land (Boone, 2000; Boone and Etchevers, 2001). The ice scheme of CARRA does not resolve the processes of snow-ice formation and internal melting of the ice, and sea ice salinity in the scheme is prescribed and constant. Surface albedo
of the sea-ice covered grid cells in CARRA is computed by applying simple parameterisation schemes. For snow-free ice cover, a temperature-dependent broadband albedo scheme is applied (defined as HIRHAM in Liu et al., 2007), and when ice is covered by snow an adapted version of the broadband snow albedo scheme by Douville et al. (1995) is used. When computing albedo of cold dry snow covering sea ice in the CARRA system, the albedo scheme of Douville et al. (1995) is modified to increase the value of the lowest possible albedo in the dry albedo degradation term from the original 0.5 to 0.75. Sea ice albedo
schemes applied in CARRA do not explicitly distinguish between direct and diffuse components of surface albedo (sometimes referred to as black-sky and white-sky albedo, see, e.g., Lucht et al. (2000) for additional details) and compute model albedo based only on the state of the ice surface without taking into account atmospheric state or such variables as solar zenith angle. For the open-ocean part of a grid cell, the albedo scheme of Taylor et al. (1996) and a constant albedo of 0.06 are used as direct and diffuse albedo components, respectively. Finally, the grid-cell average albedo of a sea ice grid cell is computed as a
weighted by the ice concentration mean of sea ice and open sea albedos (when the land fraction within a grid cell is not zero, further weighted averaging is performed to incorporate the land surface albedo). However, the HARMONIE-AROME NWP system does not produce grid-cell averaged albedo as an output variable, therefore in the released CARRA product the surface albedo field is a diagnostic computed from the hourly accumulated downwelling and upwelling shortwave radiation fluxes and available only from the model integration output.

The CARRA system is based on a classic non-coupled regional short-range NWP system that does not include a prognostic ocean model nor a slab mixed layer parameterisation scheme. The sea surface temperature and ice concentration fields in CARRA are prescribed from observational data sets and, as a result, the sea ice scheme can not freeze new ice during the model integration. Therefore, sea ice extent in the CARRA system is updated only at the analysis time by means of updating the ice concentration field and using a simple extrapolation-like procedure for initialisation of the prognostic variables of the sea
ice scheme. This new ice, placed in previously ice-free grid cells, is always snow-free and snow cover over sea ice in CARRA is accumulated during the model forecast from the model precipitation. In cases when ice concentration is adjusted from one non-zero value to another non-zero value within a grid cell, both ice thickness and snow depth remain unmodified (thus, the snow volume is not conserved when ice concentration increases in a grid cell). The following satellite sea ice concentration products are used in CARRA over the Arctic Ocean: ESA CCI SICCI (Toudal Pedersen et al., 2017), which is applied whenever
available, and OSISAF OSI-450 (Tonboe et al., 2016) as a fallback data set to be used when ESA CCI data are missing (Yang et al., 2020).

Over the ice-covered grid cells the CARRA system does not apply any surface data assimilation or relaxation procedure, thus the sea ice model is not constrained by observations (except for prescribing the sea ice concentration from an external data set). At the initial cold start of a reanalysis production stream the system is initialised with snow-free ice cover with a uniform thickness of 0.75 m and the temperature set to the freezing point of the sea water. Then, a one year spin-up period is used for preparing the initial model state in that CARRA production stream. This spin-up period was deemed practically-sufficient for the reanalysis production, however it can not eliminate discontinuities in slowly varying unconstrained variables, such as ice thickness for grid cells with multiyear ice cover (see Appendix A for more details).

Coupling between the ice surface and the model atmosphere follows the original implementation of Batrak et al. (2018), however, form drag over sea ice is not taken into account in the CARRA system.

## 2.2 ERA5

ERA5 (Hersbach et al., 2020) is a fifth-generation global atmospheric reanalysis product developed by the European Centre for Medium-Range Weather Forecasts (ECMWF), which covers the time period from 1950 to the present (2023, at the moment of writing this manuscript). The ERA5 reanalysis system is based on the ECMWF 4D-Var data assimilation and forecasting system (IFS-HRES), and provides data on a reduced Gaussian grid with a nominal horizontal resolution of 31 km.

Sea ice cover in the ERA5 system is modelled in a similar to CARRA way, by using a one-dimensional sea ice parameterisation scheme, although the sea ice model of ERA5 is considerably simplified compared to that of CARRA. In ERA5, sea ice has a constant and uniform thickness of 1.5 m and it does not apply an explicit prognostic parameterisation of the snow cover. Sea ice concentration in ERA5 is also provided from an external source (several data sets are used throughout the time period covered by the ERA5 product, see Hersbach et al. (2020) for additional details) and not modified by the modelling system during the model integration. Surface albedo of the sea ice cover in the ERA5 system is represented by time-interpolated monthly values of Ebert and Curry (1993). For winter months the dry snow albedo is used to simulate the effects of snow cover in the snow-free parameterisation scheme (ECMWF, 2016).

## 3 Observational data sets and methods

To assess the performance of CARRA in representing Arctic sea ice cover, we validate the model output against a wide set of remote-sensing and observational products as well as in situ observational data sets. In this section we provide a summary of the applied processing methods and an overview of the utilised data sets.

### 3.1 General design of the validation procedure

Where applicable, we use verification scores computed for both CARRA and ERA5 to show the performance of the new regional reanalysis product as compared to the global one. However, ERA5 has much lower spatial resolution than CARRA (for the sake of convenience, in the present study we use ERA5 data interpolated from the native reduced Gaussian grid to a 0.25° regular latitude-longitude grid, which does not result in any significant information loss but greatly simplifies the

further data processing). Thus, when computing scores based on high-resolution observational data sets (for example, for the sea ice surface temperature) for the ERA5 reanalysis we resample the data from the regular 0.25° grid onto the 2.5 km CARRA grid using nearest neighbour interpolation. After applying such a procedure the ERA5 fields on the CARRA model grid still represent the variability of the original ERA5 data set thus potentially degrading some of the verification scores due to oversampling when using observational data sets with high spatial resolution. However, impacts of this oversampling on ERA5 scores are not assessed in the present study and computed scores are used without further correction when comparing performance of ERA5 and CARRA. On the other hand, when using coarse-resolution products, such as satellite ice thickness retrievals, both CARRA and ERA5 are aggregated on the product grid as a first step before computing verification scores.

The model domains of CARRA include considerable parts of the European and Canadian Arctic and the characteristics of sea ice cover vary in different parts of the area represented in CARRA. For example, Baffin Bay, which is locked between Baffin Island and Greenland, and connected to the Central Arctic by a few straits, is primarily covered by first-year ice and has a low amount of multiyear ice transported from the Central Arctic (Tang et al., 2004; Dunbar, 1973). In contrast, sea ice cover of the Greenland Sea includes a considerable amount of old multiyear ice exported through the Fram Strait by the East Greenland Current (Aagaard and Coachman, 1968; Schmith and Hansen, 2003). The Barents Sea, unlike Baffin Bay, is not locked between land masses and is better connected with the central Arctic Ocean, thus it has a very dynamic ice cover (Vinje and Kvambekk, 1991). Therefore, in the present study we assess performance of CARRA in representing the ice conditions for a selected set of regions in addition to verifying the performance of the system over the whole ice-covered part of the model domain. The following four areas of interest are introduced (see Fig. 1): zone A – Baffin Bay (including the Nares Strait) and Davis Strait; zone B – Greenland Sea and the part of the North Atlantic Ocean adjacent to the Greenland coast; zone C – Barents Sea, Kara Sea, White Sea; zone D – central part of the Arctic Ocean within the CARRA domains defined by the northern borders of zones A, B and C. Borders of the zones A–D are set following the definitions of sea boundaries by IHO (International Hydrographic Organization) (1953) complemented by the proposed by the IHO boundaries of the Iceland sea for the sake of convenience.

To study the long-term evolution of sea ice in the CARRA product, in addition to validation against observational products, we assess the series of mean monthly anomalies of sea ice surface temperature, ice thickness and snow depth. The anomalies are computed against reference multiyear mean fields constructed using the CARRA data over a 20 year time period from 2000 to 2020. The reference period of 20 years was selected to allow comparisons of the sea ice surface temperature anomaly trends in CARRA to those derived from an observational product, which is not available prior to 2000.

The large data volumes of the CARRA and ERA5 products often do not allow for the direct computation of quantiles of a parameter of interest due to limitations of the processing hardware. Thus, when direct computation is not feasible, we use the algorithm suggested by Greenwald and Khanna (2001) to compute quantiles which, while not being mathematically precise, are accurate enough for the purposes of the present study. In the text we explicitly distinguish between approximate and precise quantiles by using the term 'estimated quantiles' for the former case.

## 3.2 MODIS ice surface temperature products

The majority of the atmospheric reanalyses are based on adapted versions of operational NWP systems, and sea ice in these products is often represented by simplified one-dimensional sea ice parameterisation schemes. Ice surface temperature is one of the most important parameters in such schemes since ice cover is treated as a lowest boundary condition for an atmospheric model of an NWP system and not as one of the main prognostic components of the system. Other parameters such as ice thickness, snow depth or snow-ice interface temperature, while undoubtedly important to accurately represent the evolution of ice cover and valuable for end users of a reanalysis product, do not *directly* affect the energy exchange between the ice surface and the model atmosphere, thus their quality (as long as the produced surface temperature is realistic) is less critical to the reanalysis system itself.

In situ observations are local and sparse in the Arctic, thus, to obtain a general overview of the quality of ice surface temperature in CARRA, remote-sensing products are employed as the main source of observational data. In the present study we use near real time (NRT) level-2 (Parkinson et al., 2006) ice surface temperature products based on data from the MODIS instrument onboard the Terra and Aqua satellites (Hall and Riggs, 2015a, b). The MODIS sea ice surface temperature product is provided in 5 minute swathes, which have a nominal resolution of 1 km. Since the product is based on data from infrared-sensitive channels of the instrument it provides estimates of the ice surface temperature only in cloud-free conditions. Therefore, MODIS retrievals of ice surface temperature tend to have a cold bias when compared to in situ observations (Hall et al., 2004; Herrmannsdörfer et al., 2023; Li et al., 2020). Moreover, errors in cloud-detection, which can be challenging over sea ice, would result in spurious 'cold' pixels in the product. Nevertheless, the MODIS product has been shown to provide ice surface temperature fields of high enough accuracy (Hall et al., 2004) for the purposes of the present study. Additionally, the MODIS data record covers a considerable time period allowing for assessing the multiyear performance of the reanalysis products without employing multiple retrievals based on data from different satellite instruments which simplifies intercomparisons and analysis.

For the intercomparisons we use the MODIS product data sets from both Terra and Aqua satellites covering the period from 2000 (Terra, the product from the Aqua satellite is available from 2002) to 2020. When processing, the two observational products are treated as a single merged data set and referred to as the MODIS ice surface temperature product in the following text. To reduce the impact of the misrepresented pixels of the MODIS product, we select only the pixels marked as 'good quality' by the quality assessment procedure of the ice surface temperature retrieval algorithm. To compare gridded reanalysis fields against the non-projected satellite product, MODIS data are aggregated on the CARRA model grid (separately for the two CARRA model domains). When aggregating for a selected valid time of a reanalysis product all the MODIS swathes within the $[-30; 30)$ min interval are used without any time interpolation or adjustment. When comparing model data against the MODIS product, only the cloud-free sea ice grid cells of a reanalysis product (total cloud cover is less than 0.125) with ice concentration over 15% are used.

### 3.3 Satellite albedo products

Similarly to the ice surface temperature, sea ice albedo is an important parameter which has a strong effect on the surface energy budget through the albedo feedback mechanism: a decreased albedo leads to more absorbed radiation, which again leads to higher surface temperatures and loss of sea ice (Riihelä et al., 2021; Pistone et al., 2014; Curry et al., 1995). Therefore, to assess the modelled albedo fields in CARRA we compare them to a satellite-based surface albedo product. Additionally we perform a similar comparison using ERA5 to study the potential added value of the regional reanalysis product.

In the present study we use the surface albedo product (SAL) of the CLARA-A2 data record, a 34-year time series of black-sky surface albedo (covering the time period from 1982 to 2019), which is based on Advanced Very High Resolution Radiometer (AVHRR) data from the polar orbiting NOAA and METOP satellites (Karlsson et al., 2017). This product has been validated against in situ observation in earlier studies (see, e.g., Karlsson et al., 2017; Anttila et al., 2016) and it is known to perform reasonably well over sea ice. Technically, SAL is provided on a 25×25 km equal-area grid (over the polar regions) and it is available as monthly or 5-day means.

For the albedo comparison we selected a 15-year time period, 2000-2015 (thus, avoiding the extension part of the CLARA-A2 data record), which reflects modern sea ice conditions of the Arctic well. To match the monthly means of the SAL product, we perform similar averaging of the hourly output data from both CARRA and ERA5 over each month from April to September (for other months the SAL product does not provide enough observations over the study area due to insufficient light conditions). Only the product grid cells with the SAL monthly means derived from over 100 valid clear-sky AVHRR observations at global-area coverage resolution (4 km) are included in the analysis. Both CARRA domains are included in the intercomparison, and for the overlapping region an average of the albedo fields from both domains is used. To compare the albedo from the coarser-gridded SAL to the reanalyses, we aggregate CARRA and ERA5 albedo fields in the product grid. Additionally, for CARRA we consider only the aggregates with at least 40 CARRA grid cells within a SAL product pixel, and for ERA5 with at least 2 grid cells. For both reanalyses this extra check ensures that no less than a half of the SAL grid cell area is represented, without extending to the adjacent grid cells. Finally, the monthly mean error is computed for each grid cell of the aggregated reanalysis fields.

### 3.4 Satellite sea ice thickness retrievals

The evolution of sea ice thickness is not the main target process of the one-dimensional sea ice scheme of HARMONIE-AROME applied in CARRA, and, in absence of ice dynamics, it can not be reproduced with all its complexity. Nevertheless, since ice thickness is provided as one of the model parameters in the CARRA product, we compare it against an observational product to highlight the limitations of the produced data set.

For this task, similarly to ice surface temperature and ice albedo, we use a satellite product to obtain a considerable spatial and temporal coverage of sea ice within the area covered by CARRA. Specifically, we utilise a weekly combined CryoSat-2 and SMOS product (Ricker et al., 2017), which uses satellite altimetry data from the CryoSat-2 satellite for estimating the thickness of thick ice while taking estimates based on passive-microwave observations taken by the SMOS satellite over thin

ice. The product covers the time period from 2010 to 2021 (at the moment of writing this paper) and it is provided on a 25 km grid. Data gaps in the estimated weekly ice thickness fields are filled by means of an optimal interpolation procedure where the background field is produced by merging past and future (relative to the valid time of the produced analysis field) estimates derived from CryoSat-2 and SMOS (Ricker et al., 2017). Due to the limitations of the retrieval algorithms, the product does not provide ice thickness estimates between May and October.

When comparing the ice thickness reported by CARRA against the observational product, the three-hourly CARRA fields are aggregated on the 25 km grid of the product and then weekly-average values are computed. Over the overlap area of the western and eastern CARRA model domains, the final ice thickness within a grid cell is computed as a mean of the values obtained from the two domains. Additionally, to assess the potential added value of applying a thermodynamic sea ice model in the CARRA system we use scores computed from the uniform value of 1.5 m of ERA5 as a baseline.

### 3.5 Satellite snow depth over sea ice retrievals

The quality of snow cover in the CARRA product is of higher interest than that of ice thickness since misrepresented snow cover can result in larger errors in the modelled surface energy balance compared to the effects induced by errors of a similar scale in a misrepresented ice layer beneath the snow cover. However, satellite-based snow depth retrievals are much more uncertain and less reliable compared to ice thickness retrievals. Moreover, drifting ice mass balance buoys, which can be a valuable source of in situ observations tend to enter the area covered by the CARRA model domains in the spring time when snow cover starts actively melting thus provide little insight into the snow accumulation period.

Thus, in the present study we use a satellite-based snow depth product (Lee et al., 2021) for assessing the snow depth in CARRA and complement it by the Operation IceBridge (Sec. 3.6) flight campaign retrievals. The utilised product is based on the algorithm of Shi et al. (2020) where monthly estimates of the snow depth over sea ice are computed using sea ice freeboard derived from passive-microwave data. The applied algorithm uses monthly surface and snow ice interface temperature fields for estimating the snow depth to ice thickness ratio, which is, in turn, used to derive the snow depth from the estimated freeboard. The product covers the time period from 2003 to 2020 and provides pan-Arctic fields of the estimated snow depth on a 25 km grid for January, February, and March.

Similar to the ice thickness fields, snow depth over sea ice in the CARRA product is aggregated on the 25 km grid of the product and monthly means are computed. The region where western and eastern CARRA domains overlap is treated in the same way as when assessing the ice thickness.

Additionally, when comparing the CARRA data against the observational product, monthly CARRA snow depth 'estimates' are computed by applying the algorithm of Shi et al. (2020) to the model freeboard obtained by using the actual snow and sea ice parameters (i.e. the model snow water equivalent, model snow density and ice thickness). Applying the same algorithm as was used in the product to the model data allows highlighting the discrepancy between the model snow depth in CARRA and the product arising from the differences in the snow and ice parameters. The snow depth in CARRA retrieved using the Shi et al. (2020) algorithm is referred to as 'corrected snow depth' later in the text.

Since the ERA5 reanalysis system does not resolve the evolution of the snow cover over sea ice, ERA5 was excluded from the snow depth intercomparisons.

### 3.6 Operation IceBridge snow depth data

Since the satellite snow depth retrievals tend to have high uncertainty, we use an additional independent data set to complement the comparisons against the satellite product. In the present study we use snow depths obtained from the radar altimetry observations taken during the Operation IceBridge (OIB) flights (Kurtz et al., 2015, 2016). This data set spans over the time period from 2009 to 2019 and has uneven spatial coverage with most of the flights within the CARRA domains conducted over the north of Greenland and only few tracks entering the areas south of 80°N.

To compare the CARRA snow depth against the OIB data, the snow depth estimates along the OIB flight tracks, which have a spatial resolution of 40 m, are aggregated on the 2.5 km grid of the CARRA product. For intercomparisons, snow depths from the CARRA analysis with the closest valid time are considered for each data point of the aggregated OIB track.

### 3.7 In situ data and ice mass balance buoys

Satellite retrievals discussed so far in the previous sections provide estimates of the sea ice properties in the Eulerian frame, or in other words, over a prescribed grid. Thus, for these products changes in sea ice state within each grid cell arise due to contributions from both thermodynamic and dynamic processes. However, the CARRA system uses a greatly simplified sea ice parameterisation scheme which represents only thermodynamic processes in the ice column. Therefore, to better assess the performance of the CARRA system in representing these processes we compare the CARRA product against a set of in situ observations reported by drifting ice mass balance buoys.

The unmanned ice mass balance buoys (IMB) usually measure snow depth, ice thickness and temperature and can vary in design and complexity. In the present study we use data from two types of IMBs: acoustic sounder-based buoys (Richter-Menge et al., 2006), referred to as CRREL buoys in the text, and simpler thermistor string-based buoys (Jackson et al., 2013), referred to as SIMBA buoys in the text. The CRREL IMBs measure the distances between the downward-looking sounder and the snow surface, and the upward-looking sounder and the ice bottom. Based on initial sea ice conditions at the time of buoy deployment, these distances can be converted to snow depth and ice thickness. Additionally, CRREL buoys employ a separate thermistor string that measures the vertical temperature profile through air-snow-sea ice-ocean. The thermistor string of CRREL buoys has individual sensors located at a distance of 10 cm between each one. The SIMBA IMBs measure only the series of vertical temperature profiles by means of a thermistor string with sensors located every 2 cm. However, two types of temperatures are measured by the SIMBA buoys. Firstly, they report the environment temperature of air, snow, ice and water where SIMBA thermistor sensors are located, which is consistent with the temperature profiles reported by CRREL buoys. Secondly, SIMBA buoys measure the temperature change after each thermistor is applied with an identical amount of heat by means of heating elements adjacent to the sensors. The changes in the temperature reading after a heating cycle depend on the thermal properties of air, snow, ice and water, and therefore can be used to identify the type of medium surrounding the sensors. Thus, temperatures reported by SIMBA buoys can be used to derive snow depth and ice thickness manually (Lei et al.,

2018) or automatically (Liao et al., 2018; Cheng et al., 2021). Both types of IMBs are normally deployed on undeformed ice floes at a selected location and then drift along with the ice floe. The standard observations are made every 6 hours, and the buoy's GPS location is recorded every hour. Both types of IMB have been deployed in the Arctic Ocean for many years. Their data are representative for regional, seasonal and interannual sea ice mass balance (Perovich and Richter-Menge, 2015; Lei et al., 2018) and air-sea ice-ocean interactions along IMB drift trajectories (Provost et al., 2017; Koo et al., 2021; Cheng et al., 2021; Lei et al., 2022). In this study, we use data from 19 individual IMBs (see Table S5) collected from various field programs and compare them against the CARRA product (using ERA5 as a baseline, where applicable). Other IMBs, which also entered the CARRA domains throughout the time period covered by the product, were excluded from the intercomparisons due to issues with the reported parameters. We target the following four parameters: snow depth, ice thickness, surface temperature and snow-ice interface temperature. The surface temperature was obtained by linear interpolation based on snow depth (or ice thickness) and readings from the thermistor sensors closest to the snow-air interface (or ice-air interface in case of missing snow cover). The snow-ice interface is assumed to remain unchanged from its initial position when an IMB was deployed. Although, dynamic and thermodynamic interactions between snow and ice may result in a moving snow-ice interface because of snow-ice and superimposed ice formation, especially during the early melting season and early winter when ice is still thin (Cheng et al., 2003, 2008, 2021). However, the IMBs used in this study were deployed in late autumn on thick ice floes when the ice was about to freeze up and the snow was thin, thus reducing the chances of snow-ice formation processes affecting the IMB reading. Therefore, the assumption of a static snow-ice interface is adequate for the purposes of the present study.

When processing IMB data we first identify parts of an IMB trajectory that are located within the two CARRA model domains, and a corresponding subset of observed parameters is extracted. Then, for each GPS position reported by an IMB, the selected set of parameters is retrieved from the nearest CARRA model grid cell. Note that in the CARRA product there is no dedicated snow-ice interface temperature field, therefore we used the temperature of the top-most ice layer (which can be up to 5 cm thick, see Batrak et al. (2018) for the details) as an analogue. To facilitate intercomparisons both IMB and CARRA data were resampled to hourly temporal resolution.

# 4 Results and discussion

## 4.1 Ice surface temperature

When assessing the quality of ice surface temperature in CARRA we first study the multiyear performance of the product in order to evaluate whether it reasonably represents temperature trends linked to ongoing climate change. As a second step we evaluate the annual cycle of modelling errors computed against the MODIS satellite product. Figure 2 shows the obtained anomalies as well as the computed ice surface temperature anomaly trends for both CARRA and the MODIS satellite product. As can be seen from the figure both CARRA domains show a positive anomaly trend with a value of 0.08 $\,^\circ\mathrm{Cy}^{-1}$ (95% CI [0.06; 0.11] $\,^\circ\mathrm{Cy}^{-1}$) and of 0.20 $\,^\circ\mathrm{Cy}^{-1}$ (95% CI [0.16; 0.25] $\,^\circ\mathrm{Cy}^{-1}$) for the western and eastern CARRA model domains, respectively. Monthly sea ice surface temperature anomaly trends found in the MODIS product show comparable values for both CARRA model domains; 0.07 $\,^\circ\mathrm{Cy}^{-1}$ (95% CI [0.05; 0.10] $\,^\circ\mathrm{Cy}^{-1}$) for the western domain and 0.17 $\,^\circ\mathrm{Cy}^{-1}$ (95% CI

[0.13; 0.21] $°Cy^{-1}$) for the eastern domain. These values are in line with the findings of previous studies (see, e.g., Rantanen et al., 2022; Nielsen-Englyst et al., 2023), but differences in the lengths of the anomaly series and covered areas, as well as the shorter period used for computing multiyear means in the present study (20 versus 30 years), do not allow the direct comparisons. The eastern CARRA model domain showing a considerably larger anomaly trend than that found for the western domain, is also in agreement with earlier works, which suggest the Barents Sea region has higher warming rates than the

Greenland Sea and the Central Arctic region (see, e.g., Screen and Simmonds, 2010; Comiso and Hall, 2014; Isaksen et al., 2022; Nielsen-Englyst et al., 2023).

After assessing the multiyear trends in the CARRA product we focus on annual variability found in the CARRA data and on the performance of the regional reanalysis compared to the ERA5 data set. First, we assess the general performance of CARRA in terms of ice surface temperature throughout the year. Figures 3 and 4 show monthly estimated quantiles of ice surface temperature in CARRA and ERA5 compared to the estimated quantiles of the MODIS product computed for the period

from January 2000 to January 2020. The figures show that for both model domains CARRA tends to have lower ice surface temperature than ERA5 for all months, except September for the western CARRA domain where ERA5 is slightly colder. The lower temperatures of the CARRA product better agree with MODIS, especially during the winter and spring seasons over the eastern CARRA domain. During the active melting season in the summer, both CARRA and ERA5, compared to MODIS, tend

to have higher ice surface temperatures than in the retrieval and show comparable performance. With the onset of the freezing season (starting from September) and until December the two reanalysis products show a considerable warm bias similar to that found for the summer months. Moreover, during this period CARRA does not show any noticeable improvement over ERA5 for the western domain. For the eastern domain, CARRA is slightly colder than ERA5 in October and November, however it still has a higher ice surface temperature than reported by the MODIS product.

Differences in the ice surface temperature quantiles between the two CARRA model domains suggest that sea ice cover is represented with a varying degree of accuracy over the different parts of the joint area covered by the CARRA product. Thus, Fig. 5 shows the annual evolution of the ice surface temperature bias over the four selected areas of interest computed against the MODIS product for the period from January 2000 to January 2020. As can be seen from the figure, evolution of the ice surface temperature bias differs considerably over the selected areas, although CARRA still shows a lower mean error compared to

ERA5 for all zones and months except for January and mid-August to mid-October in zone A, and from mid-August to the end of September in zone B. While ERA5 almost universally has a warm bias when compared to the MODIS product (except for September in zone A) CARRA shows periods of distinct negative median bias within zone A from December to the end of March.

Over zone A, which includes Baffin Bay and the Davis Strait, ERA5 shows relatively low variation of the ice surface

temperature bias, which has a value close to 2 $°C$ with the only major drop to a value of approximately 0 $°C$ observed in September. Contrary to ERA5, CARRA has the highest positive bias in September with the value reaching 2.05 $°C$ which is reduced to zero by December and then becomes negative. A negative median bias in CARRA over zone A is found throughout the period from December to the end of March with the lower-most values of -1.66 $°C$ observed in January–February. Over the

summer season CARRA has a near-constant median bias, with values within the range of 0.87–1.45 °C, which starts growing in August.

For zone B, covering the Greenland Sea and the North Atlantic Ocean, both CARRA and ERA5 show positive median bias throughout the year. ERA5 has the highest bias in December with a value of 5.75 °C which then gradually decreases over the following months and reaches a minimum value of 1.41 °C in August before starting to grow again. For this zone CARRA shows a similar annual cycle of the median bias, although it is shifted with a maximum value of 3.63 °C observed in November and a minimum value of 0.79 °C found in July. For the period from mid-August to the beginning of October CARRA tends to have a higher positive bias than ERA5 due to a shift in the annual cycle of modelling errors.

For ERA5, zones C and D show a qualitatively similar evolution of the median ice surface temperature bias with the annual maximum observed in the autumn months and lowest bias found in July, although over zone D bias is higher on average. On the other hand, the CARRA product features noticeable differences in the annual cycle of the median ice surface temperature bias for these two zones. For zone C, CARRA, while having the highest median bias of 3.83 °C in September–October (similar to ERA5, which has a value of 4.62 °C), shows a period of the median bias reduced to near-zero from January to March (unlike ERA5 with a winter-time median bias close to 3 °C). This bias starts growing again in April to reach the summer value of approximately 1.5 °C, which is close to that of ERA5. Over zone D, CARRA does not show a winter-time near-zero median ice surface temperature bias as was found over zone C and exhibits a similar to ERA5 annual cycle with a maximum of 5.54 °C in September–October and a minimum of 0.97 °C in July. However, the autumn-time maximum of the bias, which is not present in ERA5, is well-pronounced in CARRA, similar to zone C. For zone D, the median ice surface temperature bias in ERA5 has a maximum value of 7.17 °C (November) and a minimum of 2.37 °C (July).

In general, CARRA shows the most noticeable decrease of the median bias during the winter months, when this difference can reach values of up to 4 °C, and during the melting season the difference between CARRA and ERA5 is reduced. These results, arising from including an explicit representation of the snow cover over sea ice in the CARRA systems are in line with the result of Arduini et al. (2022) assessing the effects of resolving the snow layer over sea ice in IFS-HRES. Additionally, the year-to-year variability of the ice surface temperature bias is noticeably different between CARRA and ERA5 with CARRA tending to have more variability than the global reanalysis product. This variability in CARRA is considerably higher in zones B and D than in zone A. Notably, scores over zone C in CARRA show increased variability mainly during the autumn freeze-up season, similarly to ERA5. This feature is attributed to a greatly diminished by the start of the freeze-up season sea-ice covered area within the zone C, which leads to higher contribution of areas with relatively low ice concentration, which can have considerable uncertainty, to the computed score.

## 4.2 Ice albedo

To assess the sea ice albedo fields in CARRA and ERA5 reanalysis products, we examine the monthly mean error maps for CARRA and ERA5 SALs.

Qualitatively, as can be seen from Fig. 6, sea ice albedo in CARRA and ERA5 show similar features throughout the studied months, and in both reanalyses it is higher in April, May and September compared to the satellite-based product, while in July

the sea ice albedo is underestimated. For June and August CARRA and ERA5 show noticeable difference in the sea ice albedo errors. Specifically, in June CARRA shows a positive bias in the albedo field over the northern Barents Sea, where bias is weakly negative in ERA5. In August, ERA5 shows good agreement with the CLARA-A2 product, while albedo in CARRA is overestimated in the northernmost areas of the model domain (zone D in Fig. 1).

Quantitatively, sea ice albedo in CARRA is consistently higher than values reported by the SAL product for all studied months except June and July. These high albedo values lead to larger errors than in ERA5 on average, which can be traced in the error probability density functions (PDFs, see Fig. S1). For example, in the April PDF of the albedo error, CARRA has a mode of 0.14, which is considerably higher than 0.06 found in ERA5, although these values are reduced for other months. In June, sea ice albedo errors in CARRA are distributed around zero with a mode of 0.001, but in July both CARRA and ERA5 show a clear underestimation of the albedo with the mode of the error PDF close to -0.05. In August, the error PDF of the sea ice albedo in CARRA shows a bimodal distribution which is attributed to the fresh snow accumulation and temperature drop over zone D combined with a considerable amount of snow-free sea ice grid cells still present within the CARRA model domains. As a result, the first mode of 0.08 indicates the previously observed characteristic overestimation of the surface albedo while the second mode of -0.005 is related to the snow-free ice cover within the model domain. This value of the second mode of the albedo error PDF is close to that found in the ERA5 data, namely 0.001.

In general, the observed errors in the CARRA sea ice albedo can be attributed to several factors. Firstly, sea ice albedo parameterisation schemes applied in CARRA and ERA5 systems differ in their philosophy: CARRA uses modelled albedos, while ERA5 relies on time-interpolated observation-based albedos. Therefore, in CARRA surface albedo over sea ice has more degrees of freedom and depends on the surface temperature and model precipitation. Applying an unconstrained parameterisation can result in considerable errors, even though such an approach gives a more consistent model state. Secondly, snow cover over sea ice is represented as a flat and uniform layer covering a whole grid cell with the surface albedo computed by the snow scheme. This idealised approach results in a more reflective ice surface compared to real sea ice cover. Thirdly, the negative summer-time sea ice albedo bias found in CARRA (and also observed in ERA5) indicates shortcomings in the representation of the melting regime of the sea ice. Specifically, processes of melt pond formation and evolution are not explicitly represented in the CARRA system and a simple temperature-based sea ice albedo scheme can not accurately reproduce all the effects of melt ponds. Finally, it is also possible that the SAL of the CLARA-A2 product underestimates sea ice albedo in the spring months. However, characteristic biases found in CARRA and ERA5 (for example, spring-time overestimation of the sea ice albedo) are not unique to the CLARA-A2 SAL product, and similar performance of ERA5 was observed when using other satellite based albedo retrievals as a reference (Pohl et al., 2020). Nevertheless, uncertainties in the intercalibration method of the AVHRR data record can influence the average level of the albedo, and it is expected that the upcoming next edition of the albedo product, CLARA-A3 SAL, will have somewhat higher sea ice albedo values in spring (personal communication, Aku Riihelä).

## 4.3 Ice thickness

The CARRA system is based on a non-coupled atmospheric NWP system, therefore it uses a simplified one-dimensional parameterisation scheme for representing sea ice cover in the model. However, CARRA uses a more advanced sea ice scheme compared to ERA5 and the CARRA data set includes such fields as ice thickness and snow depth. Therefore, in the present study, we use available remote-sensing and in situ observations for assessing the performance of the sea ice scheme in the CARRA system with respect to these additional parameters.

Sea ice thickness, specifically in the grid cells with perennial ice ice cover, is the prognostic model variable with longest memory in the CARRA system since it is not constrained by observations and does not disappear during summer melts, unlike snow over sea ice. Thus, consistent long-term performance of the sea ice scheme becomes more important to avoid unrealistic features in the produced data set. The long memory of sea ice becomes especially important when considering the initial production of a reanalysis data set, which represents multiple decades of data and is therefore usually generated by means of a number of separate production streams to reduce the integration time. In these streams sea ice cover is initialised independently and it can be challenging to achieve a seamless transition from one stream to another if the scheme is not constrained. Therefore, we assess the long-term performance of the CARRA system in representing sea ice cover by computing monthly mean ice thickness anomalies over the period covered by the product. Figure 7 shows the computed anomalies as well as the fitted anomaly trend for both model domains of the CARRA system. As can be seen from the figure, the CARRA product shows a general trend towards decreasing ice thickness for both model domains, namely -1.24 $\mathrm{cm\,y^{-1}}$ (95% CI [-1.38; -1.10] $\mathrm{cm\,y^{-1}}$) for the western domain, and -1.35 $\mathrm{cm\,y^{-1}}$ (95% CI [-1.48; -1.21] $\mathrm{cm\,y^{-1}}$) for the eastern domain. These values are in line with the general trend towards thinner ice in the Arctic observed and reported by multiple studies (see, e.g., Renner et al., 2014; Hansen et al., 2013; Lindsay et al., 2009), albeit with weaker thinning rates. However, Fig. 7 reveals an inconsistency in the computed anomaly series caused by separating the CARRA production into a set of parallel production streams. This inconsistency, which can be illustrated as a sudden anomaly drop between streams BE3 and S1 of the western CARRA domain as shown in Fig 7a, affects the long-term ice thickness statistics. Similar feature can be traced for the eastern domain as well, although much less apparent.

The sea ice scheme applied in the CARRA system (Batrak et al., 2018) does not resolve ice dynamics and represents only thermodynamic processes in the ice column. Thus, comparing CARRA against a gridded satellite product can highlight the limitations of the reanalysis. On the other hand, comparisons against measurements taken by drifting platforms (for example, ice mass buoys), which essentially observe only the thermodynamic processes in a single ice floe, can complement the assessment of the performance of the parameterisation scheme applied.

An initial intercomparison against the composite CryoSat-2/SMOS satellite product for all available dates shown in Fig. 9a indicates high spatial non-uniformity of the modelling errors. In general, sea ice thickness in the CARRA data set tends to be underestimated over the coast of Greenland within zone B and in the central Arctic (zone D). For other zones and areas sea ice in CARRA is thicker than reported by the satellite product. These errors in the modelled ice thickness show values between -2.2 m and 0.9 m, and are higher than the uncertainty level reported by the satellite product for most of the sea-ice

covered areas within the CARRA model domains. Similarly, non-systematic errors in the modelled ice thickness (expressed as the standard deviation of errors (ESTD), see Fig. 9b) are very non-uniform within the model domain. The highest ice thickness ESTD values are found over the Greenland Sea while over Baffin Bay and the Kara Sea CARRA shows mainly systematic errors. This distinction can be attributed to the characteristic sea ice regime of the Greenland Sea where transport of old ice from the central Arctic makes ice cover more variable and challenging to reproduce.

The annual evolution of the average mean error of the sea ice thickness modelled by the CARRA system (limited to the period of availability of the satellite ice thickness retrieval, namely from October to April) is presented in Fig. 8. As can be seen from the figure, and supported by the features found in Fig. 9, CARRA shows distinct differences in the ice thickness modelling errors and their temporal evolution between the four zones of interest. The figure shows persistent negative bias in zones B and D where the multiyear average ice thickness mean error (ME) reaches -0.88 m over zone B in January, and -1.05 m over zone D in April. For zones A and C, where ice is thinner on average, CARRA shows better performance, although the modelled ice thickness shows a positive average ME, which tends to grow throughout the winter. Thus, the ice thickness ME for zone A at the beginning of the freeze-up period is 0.07 m on average, but by mid-April it reaches values of 0.57 m. For zone C the situation is similar with the average ME ranging from 0.17 m in October to 0.52 m in April. Therefore, thermodynamic ice growth rates in CARRA over zones A and C tend to overestimate the values observed in reality. On the other hand, the zones B and D annual variability of the multiyear average ME is less pronounced. The period of ice thickness ME growth between January and April in zone A coincides with a period of negative bias in the CARRA sea ice surface temperature field (see Section 4.1), which highlights the impacts of misrepresented ice thickness on the surface energy balance in the sea ice parameterisation scheme. Notably, a similar positive ice thickness ME over zone C does not manifest in a cold bias of the sea ice surface temperature, comparable to the one observed in zone A. This discrepancy is attributed to the lower on average compactness of the sea ice cover in the Barents Sea than in Baffin Bay, which results in a higher contribution of the open-sea part of grid cells to the modelled sea ice surface temperature in zone C.

The thermodynamic sea ice model applied in CARRA is more advanced than a scheme with a prescribed and uniform ice thickness, such as that used in ERA5. To assess the improvement in the modelled ice thickness (if any) resulting from applying a thermodynamic sea ice model in a reanalysis system we compare the ice thickness ME of the CARRA product to the simulated ME of a hypothetical version of the CARRA system with a prescribed and uniform ice thickness of 1.5 m. These additionally computed scores are presented in Fig. 8. The figure suggests that having a prescribed ice thickness in the CARRA system would show reduced on average ME compared to the actual CARRA system for zones B and D but considerably increased ME for zones A and C. In general, when ice thickness is prescribed, annual series of the ME show a negative slope and the difference between the CARRA system (where ice thickness ME grows throughout the winter season or remains relatively constant) and persistent 1.5 m ice is greatest in October and reduced by April. For example, for zone D, having prescribed ice thickness would result in an October average ME of -0.05 m which is a considerably lower ME than the value of -0.78 found in CARRA, however, by the end of the winter season in April, this difference in ME is greatly reduced and ME takes values of -1.05 m and -1.03 m, respectively. For zones A, B and C the ME difference between modelled and persistent ice thickness evolves in a similar way. The reduced growth rate of the ME in CARRA compared to the persistent ice thickness indicates the

benefits of applying a thermodynamic sea ice model in the reanalysis system. However, the offset errors found over zones B and D suggest potential advantages of constraining the ice thickness by means of observational data sets.

## 4.4 Snow depth

Similarly to the ice surface temperature and ice thickness, we first assess the long term performance of the CARRA system by studying the monthly anomalies of the snow depth in the reanalysis product. As can be seen from Fig. 10, snow cover over sea ice shows a similar trend to that found for ice thickness towards more frequent negative anomalies over the last decade compared to the first half of the anomaly series. However, compared to the ice thickness, snow depth anomalies are smaller, and anomaly trends are less pronounced. Specifically, for the western CARRA model domain, the product shows a very weak negative trend of -0.09 $\mathrm{cm\,y^{-1}}$ (95% CI [-0.13; -0.06] $\mathrm{cm\,y^{-1}}$), and for the eastern model domain the anomaly trend is stronger with a value of -0.28 $\mathrm{cm\,y^{-1}}$ (95% CI [-0.33; -0.24] $\mathrm{cm\,y^{-1}}$). The general trend towards diminishing snow depth over the Arctic sea ice in both observations and modelling data sets is noted in multiple studies (see, for example, Webster et al., 2014; Chen et al., 2021; Zhou et al., 2021). The more pronounced decrease in the snow depth for the eastern CARRA model domain is in line with the modelling results of Chen et al. (2021) and Zhou et al. (2021), which show stronger negative trends in the snow depth series over the Barents Sea region.

When assessing the snow layer over sea ice in the CARRA system we use both satellite retrievals and direct observations from OIB flights. However, since the satellite retrievals of snow depth over sea ice are highly uncertain we use them only for a general qualitative assessment because they cover a much larger area compared to OIB. Figure 11 shows the mean error of the modelled snow depth in CARRA compared to the satellite retrieval product and OIB data. For both observation types CARRA shows a similar distribution of the modelling errors with generally overestimated snow depth within the model domain on average. The largest errors are found in the Greenland Sea along the eastern coast of Greenland. For this area OIB reports a snow depth of 0.24 m on average while the modelled snow depth in CARRA is 0.71 m on average. However, it must be noted that most of the OIB data in that region originate from a very limited number of flight tracks, thus the drawn conclusions should be taken with care and not generalised. For the satellite snow retrieval product a similar pattern can be traced in the Greenland Sea, which supports the aforementioned findings and suggests that snow depth is overestimated in general in the CARRA product for this region. Over the northern part of zone D, or in other words, in the part of the central Arctic present in the CARRA domains, snow depth in CARRA is considerably lower than over the Greenland Sea and, when compared to both OIB and the satellite product, shows close to zero and negative ME. Even though this ME falls below the uncertainty level reported by the satellite snow depth retrieval, consistent ME patters found in ME computed against satellite and OIB products suggest the presence of boundary effects manifesting in reduced snow accumulation along the northern boundary of the western CARRA model domain. A similar distribution of the modelling errors is observed when comparing CARRA against OIB around the Svalbard archipelago where the reanalysis product shows clear underestimation of the snow depth. This behaviour can be partially attributed to the misrepresentation errors of the sea ice cover in CARRA combined with the crude initialisation procedure for the newly ice-covered grid cells which always start from the snow-free state. However, in reality, sea ice is a drift medium and areas within and close to the marginal ice zone may contain ice floes that originate from the remote

parts of the Arctic ocean and have accumulated snow cover throughout their drift. The OIB data set does not provide snow depths over zones A (Baffin Bay) and C (Barents and Kara seas), although comparisons against the satellite retrieval product suggest that CARRA has lower snow depth modelling errors for these regions compared to the Greenland Sea. Overestimated snow depth in CARRA over zones B and D compensates to some extent the negative bias found in the ice thickness and results in a net decrease of the heat transfer between ocean and the model atmosphere, and lower ice surface temperature, in CARRA compared to ERA5.

Estimated PDFs of the snow depth in the observational products and in the reanalysis data set provided in Fig. 12 complement the findings made from assessing modelling errors. Although the differences in the PDFs computed from the satellite product and OIB data suggest that the satellite product underestimates the snow depth, as can be seen from the figure, thicker snow layers occur more often in the CARRA data than in both the satellite product and OIB observations. Modelled snow cover in CARRA has a median depth which grows throughout the winter from 0.28 m in January to 0.33 m in February and finally reaches a value of 0.37 m by March. In contrast, the satellite product reports much thinner snow cover on average with a median snow depth close to 0.13 m for all three months (see Fig. 12a). The PDFs of the corrected snow depth in CARRA show less frequent occurrence of extreme snow depths (both low and high) which indicates that the processing algorithm applied in the satellite product of Lee et al. (2021) underrepresents cases of thick snow cover within the CARRA model domains. Along the OIB tracks, median values of the snow depth in CARRA and OIB observations are 0.42 m and 0.27 m, respectively. Notably, the CARRA snow depth PDF in Fig. 12b shows a peak at zero snow depth not present in the OIB data, which suggests that there are instances of ice cover in CARRA misrepresenting the state of the actual ice cover near the ice edge, which is again attributed mainly to the effects of the applied initialisation procedure for the newly ice-covered grid cells.

## 4.5 Additional validation against in situ observations from buoys

So far in the present study we have used remote sensing data from satellite- and airborne instruments for assessing the performance of the CARRA system in reproducing the evolution of sea ice cover. Although providing valuable information about the sea ice state on large scales, these products rely on multiple assumptions and tend to have their own biases and limitations. Therefore we additionally assess the CARRA product using observations reported by a set of ice mass balance buoys.

Figure 13 summarises the intercomparisons between the CARRA and ERA5 reanalysis products, and ice mass balance buoy data. As can be seen from Fig. 13a, for most of the IMB trajectories CARRA report ice thickness close to or below the observed values, although there is a singe case where ice thickness is considerably overestimated in CARRA, being 0.35 m thicker, on average, than the reported ice thickness. ERA5, which has ice cover of uniform thickness, while having smaller median ME compared to CARRA (-0.08 m and -0.20 m, respectively) shows larger spread of the modelling errors throughout the set of selected buoys. Specifically, in CARRA, ME values for individual buoys range from -0.67 m to 0.35 m, and in ERA5 they are within the interval from -0.78 m to 0.61 m. Such a discrepancy in the performance of the two reanalysis products suggests that the thermodynamic sea ice scheme applied in CARRA, while having obvious deficiencies, is more suitable for representing the evolution of a drifting ice floe. The standard deviation of errors in the modelled ice thickness in CARRA is higher of that

in ERA5, which is in line with the increased complexity of the sea ice model in the regional reanalysis system (in ERA5 ice thickness is prescribed and constant, thus computed ESTDs simply represent variability within the observational series).

The snow depth over sea ice in CARRA, when compared to buoy data, also shows similar variability between individual buoys, although the median value of the per-buoy ME is positive and has a value of 0.10 m, while errors from individual buoys range from -0.34 m to 0.34 m. This result is in line with the findings of earlier intercomparisons against the satellite snow depth product and OIB observations where CARRA shows consistent overestimation of the snow depth in the area. However, there is a substantial number of buoys (5 of 21 in total) that report a considerable underestimation of snow depth in CARRA, with along-track ME below -0.10 m. For four of these five buoys, underrepresented ice cover in CARRA can be partially attributed to the offset error, as these buoys report a snow layer that is considerably thicker than in CARRA already at the beginning of the buoy's trajectory section within the CARRA model domains (this initial offset error takes values from 0.14 to 0.79 m). The last buoy in this group enters the western CARRA domain during the melting season 2012 and shows much faster snow accumulation compared to the reanalysis product during the following autumn months. This discrepancy in snow depth between buoy data and CARRA is attributed to boundary effects in the regional reanalysis system, which has a pronounced lateral spin up zone for precipitation (see Appendix B for additional details).

In the presence of offset errors, correlation can be used as an additional indicator of the level of agreement between the observational and modelled values. However, for the discussed set of ice mass balance buoys, computed values of the correlation coefficient show considerable variation ranging from strong correlation to strong anticorrelation for both snow depth and ice thickness (see Table S5 for the actual values). Such a discrepancy between the modelled and observed parameters can be partly attributed to boundary effects near the edge of model domain and to the crude procedure of the initialisation of new ice.

Ice and snow temperature series reported by buoys are a valuable source of in situ observations, which can be used to assess the performance of a reanalysis system, especially for the parameters that are not available from the satellite products, such as the snow-ice interface temperature. Figure 13d shows estimated PDFs of the modelling errors of the ice surface temperature in ERA5 and CARRA reanalysis products, as well as the PDF of the snow-ice interface temperature errors in CARRA. As can be seen from the figure, CARRA shows a lower probability of positive errors in the modelled ice surface temperature than ERA5 and a higher probability of the modelled ice surface being colder than reported by buoys. Additionally, the mode of the CARRA ice surface temperature error PDF is -0.19 °C, which is closer to zero than in ERA5, where the mode has a value of 0.62 °C. These error distributions support the conclusions drawn from comparing the reanalysis products against the MODIS ice surface temperature product. Similar to those results, CARRA, when compared to buoys, tends to show lower than ERA5 surface temperatures with ice surface temperature MEs of 0.04 °C and 1.48 °C, respectively. Unlike the ice surface temperature, the snow-ice interface temperature, when modelled by the CARRA system, shows a notable warm bias with the error PDF having a mode of 0.77 °C. The positive bias in the snow-ice interface temperature found in the CARRA data (ME is 0.61 °C) is attributed to the commonly occurring overestimation of the snow depth found when comparing CARRA against satellite and OIB snow depth, which is also identifiable when comparing reanalysis data against IMBs (see Fig. 13a). In such cases, the insulating effect of the snow layer is too strong, which leads to a higher snow-ice interface temperature, especially if the ice thickness is also underestimated. When assessing the modelled and observed temperature series for individual buoys,

CARRA shows consistently high correlation with the buoy data for both surface and snow-ice interface temperatures. For surface temperature, the correlation coefficient ranges from 80% to 97%, while for the snow-ice interface temperature the correlation is slightly lower on average with the lowest value of 63% (see Table S5).

Since the main purpose of the sea ice parameterisation scheme in the CARRA system is representing the heat exchange between the ice and the model atmosphere, we additionally assess the heat fluxes throughout the snow and ice layers as they govern the heat transport from the ocean to the atmosphere. Figures 13b and 13c show the temperature gradient within the ice and snow layers, respectively, as reported by the CARRA product compared against the values computed from the buoy data. The figures show good agreement between the modelled and observed gradient in most cases, although CARRA tends to underestimate the highest values of the ice temperature gradient compared to buoy data, which can be attributed to the warm bias of the snow-ice interface temperature. One notable exception is a single CRREL buoy deployed in 2010 for which CARRA reports much higher temperature gradients within the ice layer than observed as the buoy drifts within the CARRA model domains. For this buoy, CARRA shows a considerable underestimation of the ice thickness at the point where the buoy entered the model domain. This initial error of $0.75$ m in the modelled ice thickness resulted in an overestimated temperature gradient in the ice layer with a maximum value of $0.19\ {}^{\circ}\mathrm{C}\ \mathrm{cm}^{-1}$ compared to $0.06\ {}^{\circ}\mathrm{C}\ \mathrm{cm}^{-1}$ as computed from buoy data. Snow temperature gradients, as shown in Fig. 13c, exhibit similar features to the ice temperature gradient in the CARRA data, although the snow temperature gradients have more spread in the computed values due to higher variability of the snow layer (primarily in terms of surface temperature) compared to the ice layer. Similarly to the ice temperature gradients, Fig. 13c suggests some underestimation of the gradient in CARRA for the strongest gradients (with a value over $0.50\ {}^{\circ}\mathrm{C}\ \mathrm{cm}^{-1}$ when computed from the buoy data). Additionally, CARRA seldomly shows negative values of the snow temperature gradient (or cases when snow ice interface is colder than the snow surface). However, some of the negative temperature gradients in the buoy data may arise from the buoys reporting positive snow surface temperatures during the melting season, thus these results should be taken with care.

## 5 Conclusions

Numerous research and engineering studies benefit from using atmospheric reanalysis products, which are sometimes treated as providing information about the *true* atmospheric state. However, since atmospheric reanalyses are generated by employing NWP systems, they are not devoid of modelling errors and biases. Moreover, a reanalysis product is produced by the same unmodified version of an NWP system that quickly becomes outdated after a few years of production. All of these factors necessitate the production of new reanalysis products that incorporate the latest developments in NWP and reflect the advances in high performance computing.

The Copernicus Arctic Regional Reanalysis (CARRA) is a novel regional atmospheric reanalysis product that focuses on the Canadian and European Arctic. This product has a considerably higher spatial resolution compared to the global ERA5 product (2.5 km versus 30 km) and is based on a non-hydrostatic regional NWP system, HARMONIE-AROME. CARRA covers the time period from 1990 to present (2023 at the moment of publication) and represents a region defined by two overlapping

model domains. Compared to ERA5, CARRA uses a more advanced sea ice parameterisation scheme, which includes explicit
representation of thermodynamic ice growth and evolution of the snow cover.

In the present study we assessed the sea ice surface temperature, surface albedo, ice thickness and snow depth fields provided
by the CARRA product and validated them against an extensive set of remote-sensing and in situ observations, with focus on
the recent decades. Additional comparisons against a selected set of IMBs complement and support initial validation against
the satellite products. Sea ice extent in CARRA was not discussed in the present study since the CARRA system uses satellite-
based sea ice concentration products, which are well-covered by earlier studies (Lavergne et al., 2019), to define ice-covered
regions.

The sea ice cover in CARRA adequately represents general multiyear trends towards thinner and warmer ice cover, con-
nected to the ongoing climate change in the Arctic. Comparisons against the satellite-based and in situ sea ice observations
show generally improved representation of sea ice in CARRA (using ERA5 as a baseline), although this improvement is not
universal. The main difference between the sea ice schemes in ERA5 and CARRA is the presence of an explicitly resolved snow
layer, which allows for much lower ice surface temperature in the CARRA system, therefore reducing the warm ice surface
temperature bias found in ERA5. However, for the area covering Baffin Bay and the Davis Strait the verification scores suggest
that a warm winter-time bias of ERA5 is replaced with a cold bias in CARRA, which is linked to overestimated snow depth and
ice thickness in these regions. This reduced in general winter-time surface temperature in CARRA can potentially benefit the
modelling system in representing stable boundary layers, as suggested by Arduini et al. (2022). An extensive validation of the
boundary layer in CARRA is out of scope of the present paper, thus we leave the assessment of the atmospheric variables over
the sea-ice covered areas in CARRA to future studies. For sea ice albedo, the CARRA product does not show improvement
compared to ERA5, which uses observation-based albedo estimates over sea ice, and displays considerable overestimation of
the spring-time surface albedo, when compared to the CLARA-A2 satellite product. This result highlights the limitations of the
unconstrained albedo parameterisation scheme used over snow-covered sea ice grid cells in CARRA and suggests that future
applications could benefit from a more detailed representation of the sea ice albedo in the sea ice model of Batrak et al. (2018),
or from constraining model albedo against an observational product. The general limitation of the sea ice scheme applied in
CARRA, namely the absence of ice dynamics and external constraints, strongly manifests itself in the verification scores for ice
thickness and snow depth. Even though, unlike ERA5, CARRA has spatially non-uniform ice thickness, it can not accurately
resolve thick multiyear ice in the central Arctic leading to a negative ice thickness bias. However, this underestimated ice thick-
ness is compensated by overestimated snow depth which results in a net decrease of the ice surface temperature in CARRA
compared to ERA5, even though the warm bias is not completely eliminated. Additionally, due to a simplified initialisation
procedure for new ice, thin first-year ice is thicker in CARRA than suggested by the reference satellite-based product. Finally,
applying an unconstrained sea ice parameterisation scheme in the reanalysis system employing several parallel production
streams results in discontinuities in the final product, which are the most pronounced in the ice thickness over multiyear ice.
Presence of such features strongly suggests benefits of constraining sea ice state by observational products in next generation
reanalysis projects. Snow cover over sea ice in CARRA exhibits similar traits related to the one-dimensional nature of the sea
ice parameterisation scheme of CARRA, such as extensive snow accumulation along the eastern coast of Greenland where

missing ice transport can not aid at removing snow-covered ice. Moreover, snow is directly affected by the boundary effects,
such as lateral spin up of model precipitation, caused by the limited area of the CARRA model domains. Thus, prognostic
snow cover over sea ice and ice thickness fields computed by the thermodynamic sea ice model of the CARRA system, which
are available within the reanalysis product and showing a reasonable annual cycle, should be used with great care.

An additional intercomparison performed against ice mass balance buoys shows good agreement between the modelled in
CARRA and observed temperatures, although the snow-ice interface temperature in CARRA has a consistent warm bias, with
700 a mode comparable to that of the ice surface temperature bias found in ERA5. Due to the location of the CARRA model
domains most of the buoys enter them at the final stage of the drift. As a consequence, ice thickness and snow depth over sea
ice show less agreement to observational series and instances of considerable offset errors were noted.

Sea ice cover in CARRA reflects current approaches applied in operational short-range applications utilising the HARMONIE-
AROME NWP system. The shortcomings and limitations of representing sea ice with non-constrained one-dimensional sea
ice parameterisation schemes, found in the present study, suggest that future generation Arctic reanalysis projects can strongly
benefit from applying more advanced approaches. For example, having a reanalysis system based on a fully-coupled NWP
system would open a possibility of representing sea ice cover as a drift medium with a much higher level of detail. However,
practical considerations might not allow applying a fully coupled atmospheric model with a strongly coupled data assimilation
system in a reanalysis project due to great computational and development costs of such a solution. Thus, constraining the state
of a simplified sea ice model by means of external data sets or data assimilation (see, e.g., Batrak, 2021; Scott et al., 2014) may
be still a viable approach to representing sea ice state in future atmospheric reanalyses.

*Data availability.*  The CARRA and ERA5 reanalysis products are available through the Copernicus Climate Data Store portal (https://cds.
climate.copernicus.eu/). The Level-2 sea ice surface temperature products from the MODIS instrument (MOD29 and MYD29) are available
from the National Snow and Ice Data Center (https://nsidc.org/). The combined CryoSat-2 and SMOS weekly ice thickness product is
715 available from the Alfred Wegener Institute (https://awi.de/). Monthly snow depth retrievals of Lee et al. (2021) are publicly available from the
authors of the original publication (https://doi.org/10.5281/zenodo.5081765). The Observation Ice Bridge snow depths data are available from
the National Snow and Ice Data Center (https://doi.org/10.5067/G519SHCKWQV6 and https://doi.org/10.5067/GRIXZ91DE0L9). CRREL
ice mass balance buoy data are available from the CRREL-Dartmouth Mass Balance Buoy Program portal (https://imb-crrel-dartmouth.org).
Data from the SIMBA buoys deployed during the N-ICE2015 campaign are available from the Norwegian Polar Institute (https://doi.org/
10.21334/npolar.2015.6ed9a8ca). Data access links for the MOSAiC SIMBA buoys, as well as for the SIMBA buoys FMI02 and FMI0606,
are provided in Table S6 and Table S7, respectively. The CLARA-A2 albedo product is available from the EUMETSAT dissemination portal
(http://doi.org/10.5676/EUM_SAF_CM/CLARA_AVHRR/V002_01).

## Appendix A:  Impact of the parallel production streams on the evolution of the sea ice variables

Operational production of the historical period of the CARRA reanalysis was performed by means of running several produc-
725 tion streams in parallel. An overview of these production streams is provided in Table A1. The CARRA modelling system does

not employ any special measures to ensure the seamless transition of the sea ice state across the production streams, which results in noticeable discontinuities. These discrepancies are most noticeable in slowly evolving variables with long memory, such as ice thickness in grid cells with perennial ice cover. Variables with a pronounced annual evolution cycle, for example snow depth or ice thickness of seasonal ice cover, show less discrepancy across the production streams because they have limited memory, and a one year spin-up period results in an adequate initial state. Therefore, in practice, ice thickness is the only sea-ice related variable which require special considerations when includes data from several production streams. As we already mentioned, for the seasonal ice cover discrepancies are small and do not have long-term consequences since the ice state in the system is discarded when a grid cell becomes ice-free. For the grid cells with multiyear ice cover the across-stream discontinuities in the CARRA model ice thickness are summarised in Fig. A1. The figure suggests that on average the median discrepancy is below 0.3 m for all production streams except S1 in both CARRA model domains. At the start of the stream S1, median discrepancy in the perennial ice thickness field, when compared to the same field at the end of the previous production stream, reaches the values of 0.83 m and 0.64 m for the western and the eastern CARRA model domain, respectively. Due to presence of such discontinuities, ice thickness of perennial ice in the CARRA product should be used with utmost care.

## Appendix B:  Boundary effects in the CARRA product

CARRA, as a regional reanalysis product is based on a limited area NWP system, therefore it relies on ERA5 data for defining the state of the atmospheric variables on the lateral boundary of the two model domains. However, differences in model physics and spatial resolution of the CARRA and ERA5 systems result in a presence of a lateral adaptation or *spin up* zone in CARRA fields. More specifically, the nesting strategy applied in the CARRA system does not utilise model level cloud water and hydrometeors from the host model, leading to a pronounced lateral spin up of the cloud cover in cases of inflow. Additionally, there is an eight grid cell-wide boundary relaxation zone for transition between the lower-resolution atmospheric state of the lateral boundaries and the higher-resolution state of the CARRA model atmosphere, where boundary effects are the most pronounced. All these effects, especially the cloud spin up, impact the model state of sea ice cover in the vicinity of model domain edges. Amongst the sea-ice related variables present in the CARRA product, snow depth is one of the most affected by the boundary effects since snow over sea ice is directly accumulated from model precipitation in the CARRA system. Due to reduced snowfall amounts near the model domain edge, snow depth over sea ice is underestimated. Figure B1 illustrates the impact of boundary effects on snow depth over sea ice in the CARRA product by comparing the model snow fields over the region of geographical overlap of the CARRA model domains. Based on this figure, when extracting sea ice variables in the region of geographical overlap we recommend considering selecting a CARRA model domain less affected by the boundary effects based on the region of interest. Outside the region of geographical overlap we anticipate the presence of boundary effects of a similar scale, however the actual size of the affected areas can not be determined due to lack of reference data. Ice thickness in the presence of boundary effects is overestimated (compared to grid cells not impacted by boundary effects) due to the reduced thickness of the insulating snow layer.

*Author contributions.* This study was devised by YB and BC. YB performed initial data retrieval and processing of CARRA and ERA5, validated ice surface temperature, ice thickness and snow depth fields against satellite retrievals, contributed to the analysis of albedo intercomparisons and to validation of CARRA and ERA5 against ice mass balance buoys. BC validated the reanalysis products against ice mass balance buoys. VKM validated sea ice albedo fields in the reanalysis products against a satellite retrieval. YB wrote the paper with contributions from BC and VKM. All authors contributed to the analysis and interpretation of the obtained results, the paper preparation, editing, and revisions.

*Competing interests.* At least one of the (co-)authors is a member of the editorial board of The Cryosphere.

*Acknowledgements.* The authors would like to thank Andrew Singleton and Morten Køltzow for their constructive comments. This work has been funded by the Copernicus Climate Change Service. ECMWF implements this Service on behalf of the European Commission. BC was supported by the European Commission, Horizon 2020 (PolarRES, grant no. 101003590). Comments and suggestions from the two anonymous referees and the editor helped to considerably improve our paper.

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

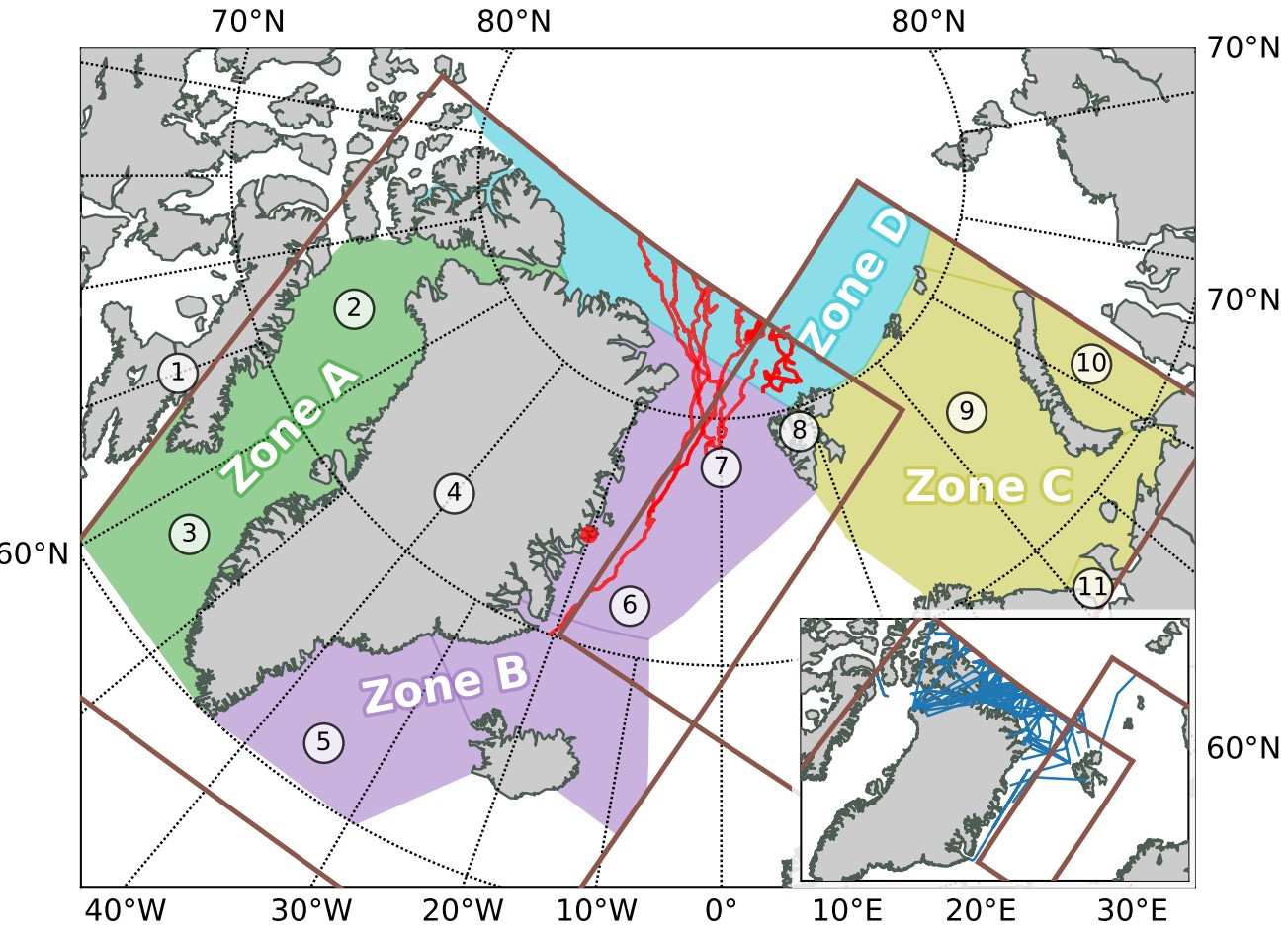

**Figure 1.** Overview of the CARRA model domains and locations of the areas of interest discussed in the present study. Also on the figure, drift trajectories of individual ice mass balance buoys (IMB) are shown (position of the IMB deployed on the land fast ice is marked with a dot). Marked on the map: 1 – Baffin Island; 2 – Baffin Bay; 3 – Davis Strait; 4 – Greenland; 5 – North Atlantic Ocean adjacent to the Greenland coast; 6 – Greenland Sea; 7 – Fram Strait; 8 – Svalbard archipelago; 9 – Barents Sea; 10 – Kara Sea; 11 – White Sea. Inset: tracks of the Operation Ice Bridge flights.

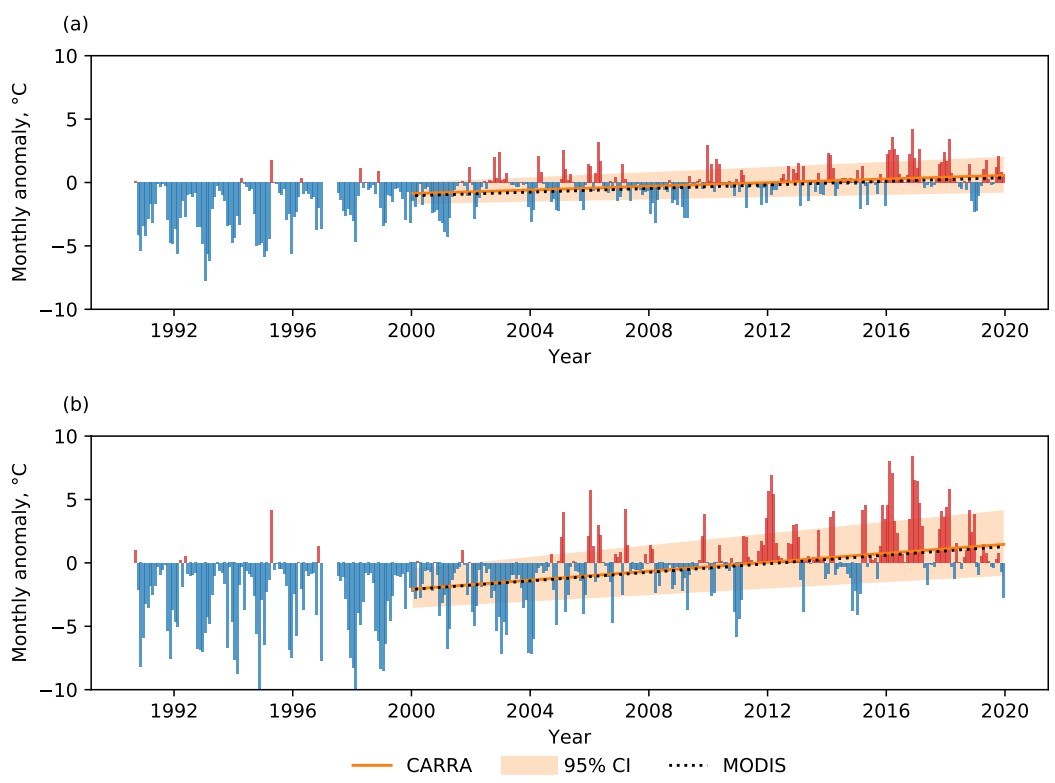

**Figure 2.** Monthly ice surface temperature anomalies in the CARRA product and fitted ice temperature anomaly trend. Multiyear monthly means computed over the time period from 2000 to 2020 are used as reference data. (a) Western CARRA model domain; (b) eastern CARRA model domain. Also on the panels, the 95% confidence interval of the CARRA anomaly trend, and the anomaly trend of the MODIS ice surface temperature product are shown.

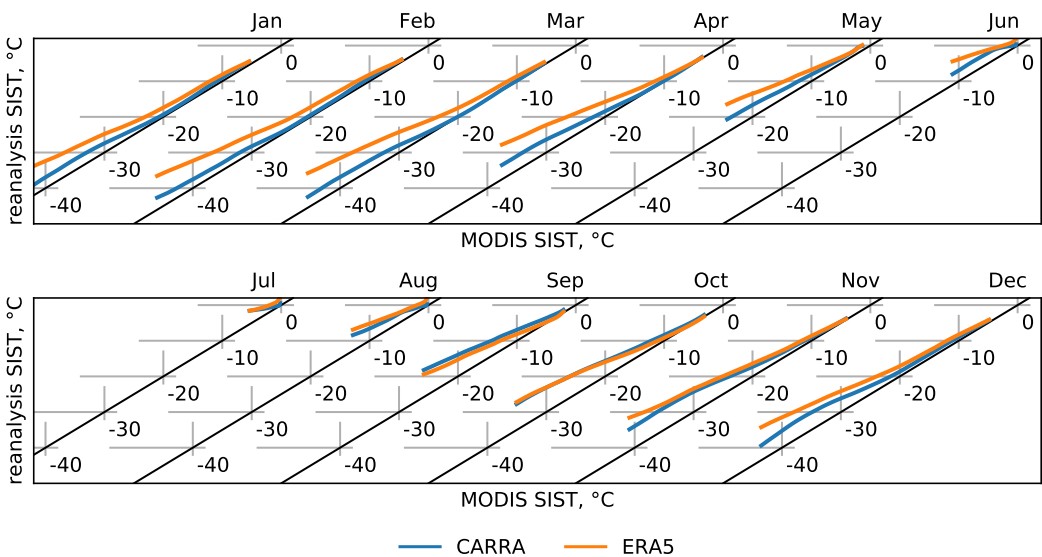

**Figure 3.** Estimated monthly quantiles of the ice surface temperature, $q \in [0.01; 0.99]$, in atmospheric reanalysis products compared against the MODIS product over the western CARRA domain. Quantiles are computed for the period from 2000 to 2020. Diagonals represent reference 1:1 match lines between observational and reanalysis quantiles. Numerical values of the computed quantiles can be found in Table S1.

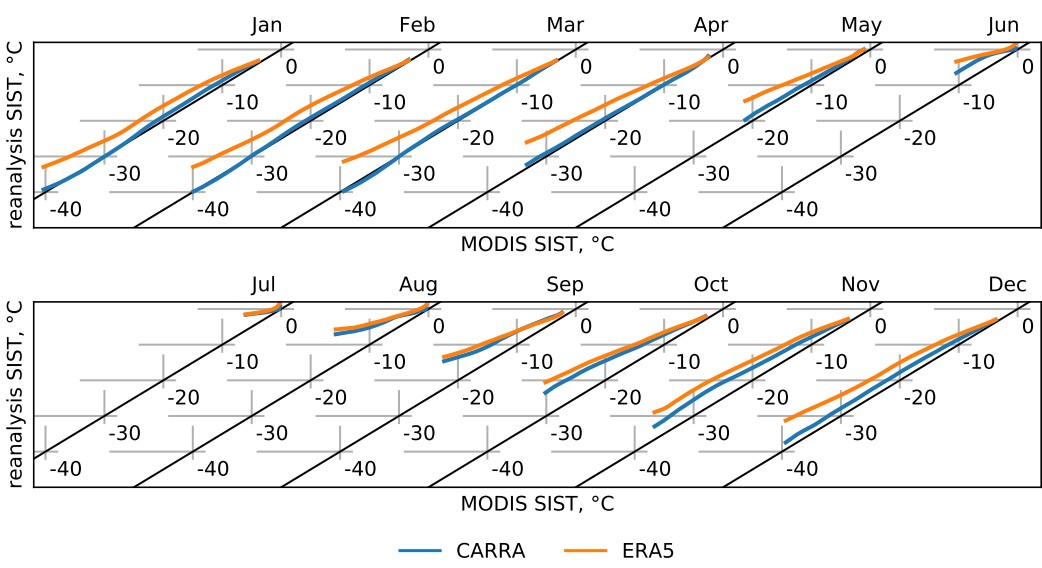

**Figure 4.** Same as Fig. 3 but for the eastern CARRA domain. Numerical values of the computed quantiles can be found in Table S2.

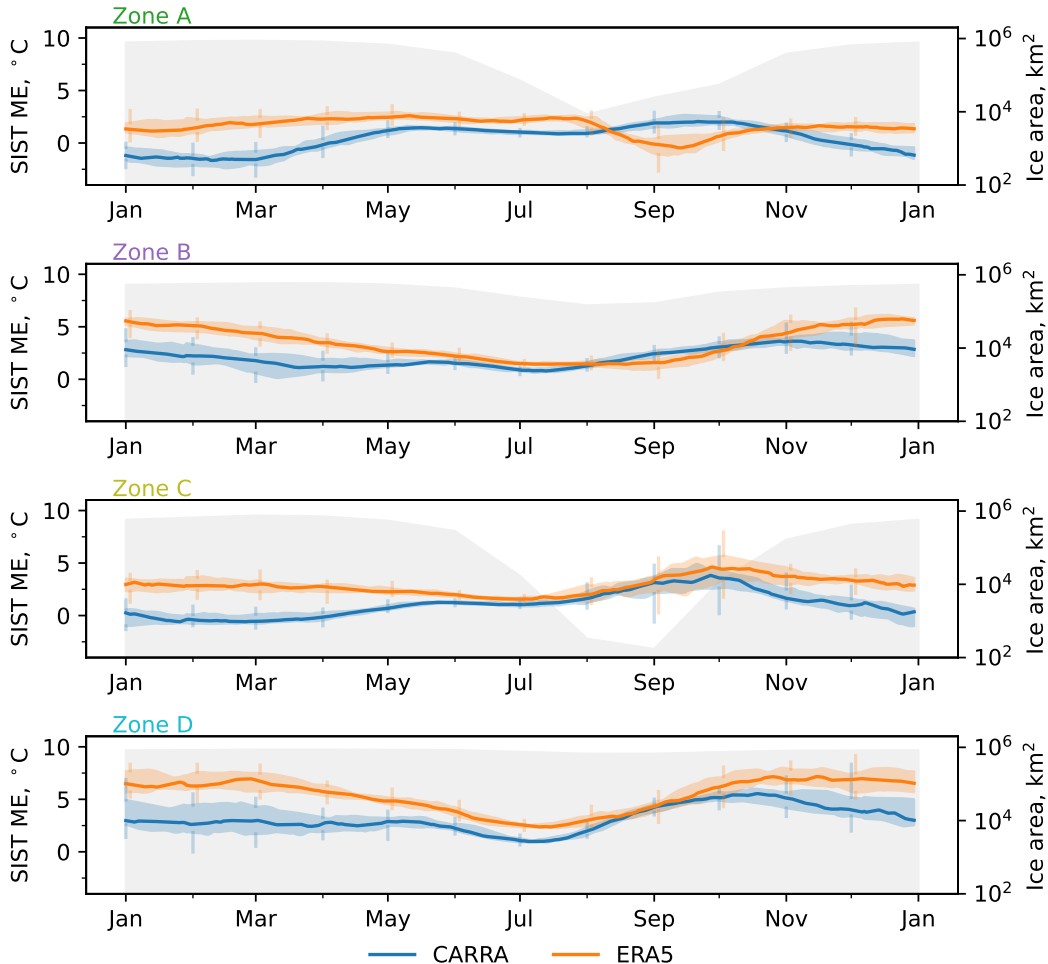

**Figure 5.** Average annual cycle of the mean error (ME) of sea ice surface temperature in CARRA and ERA5 computed over the period from 2000 to 2020 for the four areas of interest. On the panels, centre lines show the median ME value, shading bands show the corresponding interquartile ranges, and spikes show the $q \in [0.05; 0.95]$ quantile range. Values are obtained by aggregating four-week series of mean error computed against the MODIS product for each individual year in the reanalysis data sets. Numerical monthly values of the computed scores can be found in Table S3. Also in the figure, monthly median sea ice area in CARRA, computed over the same time period, is outlined.

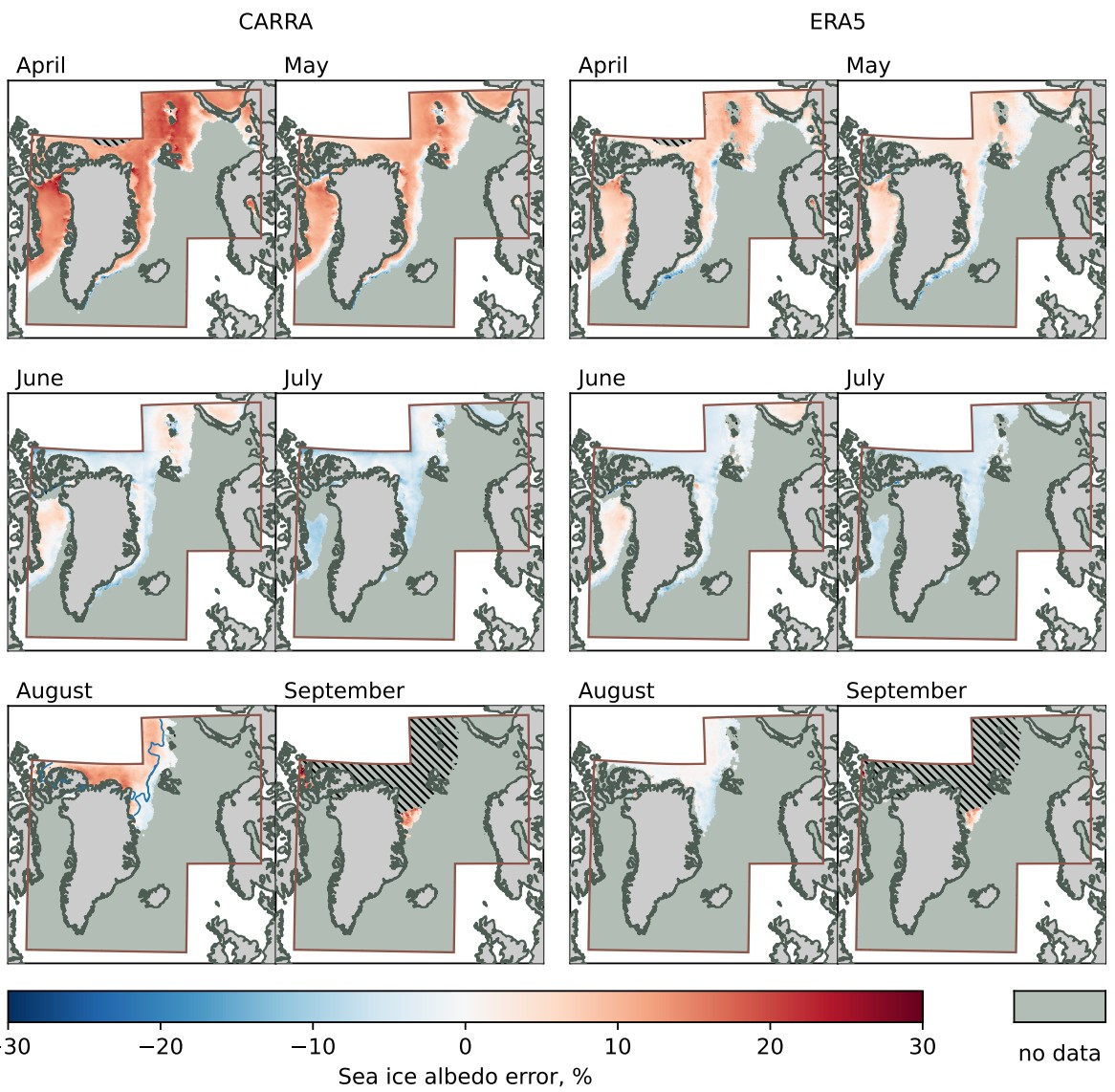

**Figure 6.** Monthly mean errors of the modelled surface albedo over sea-ice covered regions in CARRA and ERA5, computed against the CLARA-A2 SAL product over the time period from 2000 to 2015. Note that in September the observational product has considerably reduced coverage in the northern-most parts of the CARRA model domains due to challenging light conditions. Areas with missing SAL data are marked with hatches. Also in the figure, median 2000–2015 snow extent by the end of August in CARRA is outlined.

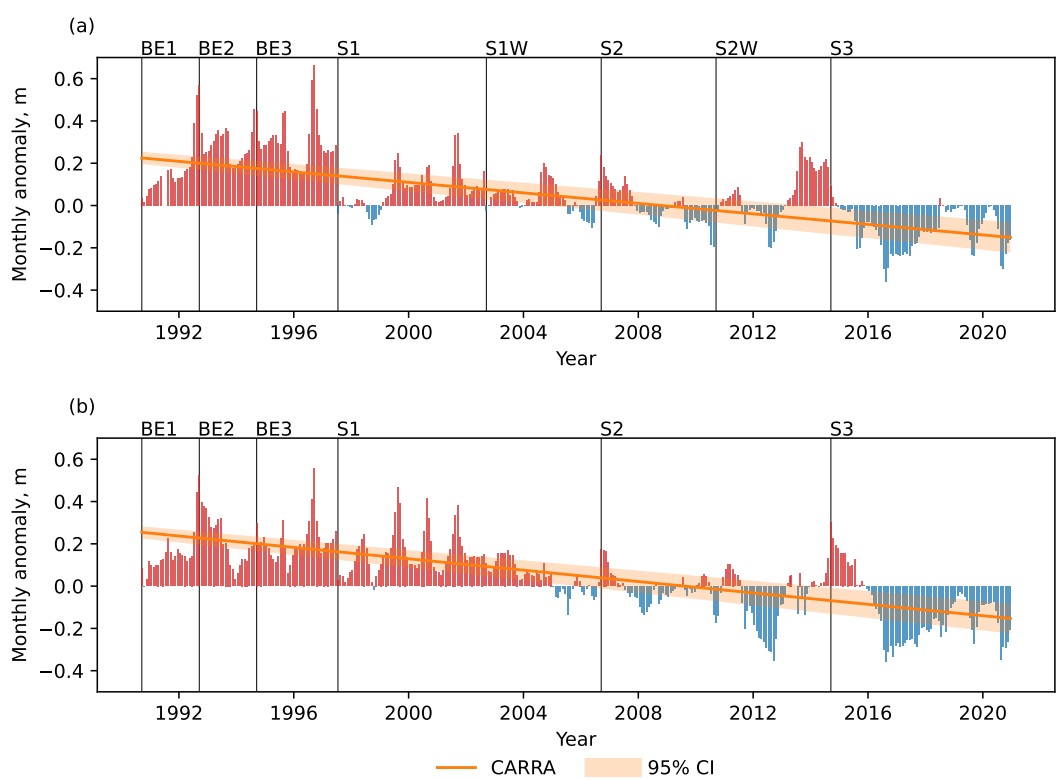

**Figure 7.** Monthly ice thickness anomalies in the CARRA product and fitted ice thickness anomaly trend. Multiyear monthly means for the time period from 2000 to 2020 are used as a reference when computing anomalies. (a) Western CARRA model domain; (b) eastern CARRA model domain. Also on the panels, the 95% confidence interval of the CARRA anomaly trend is shown, and separate productions streams S1–S5 of the CARRA system and the back extension streams BE1–3 are marked.

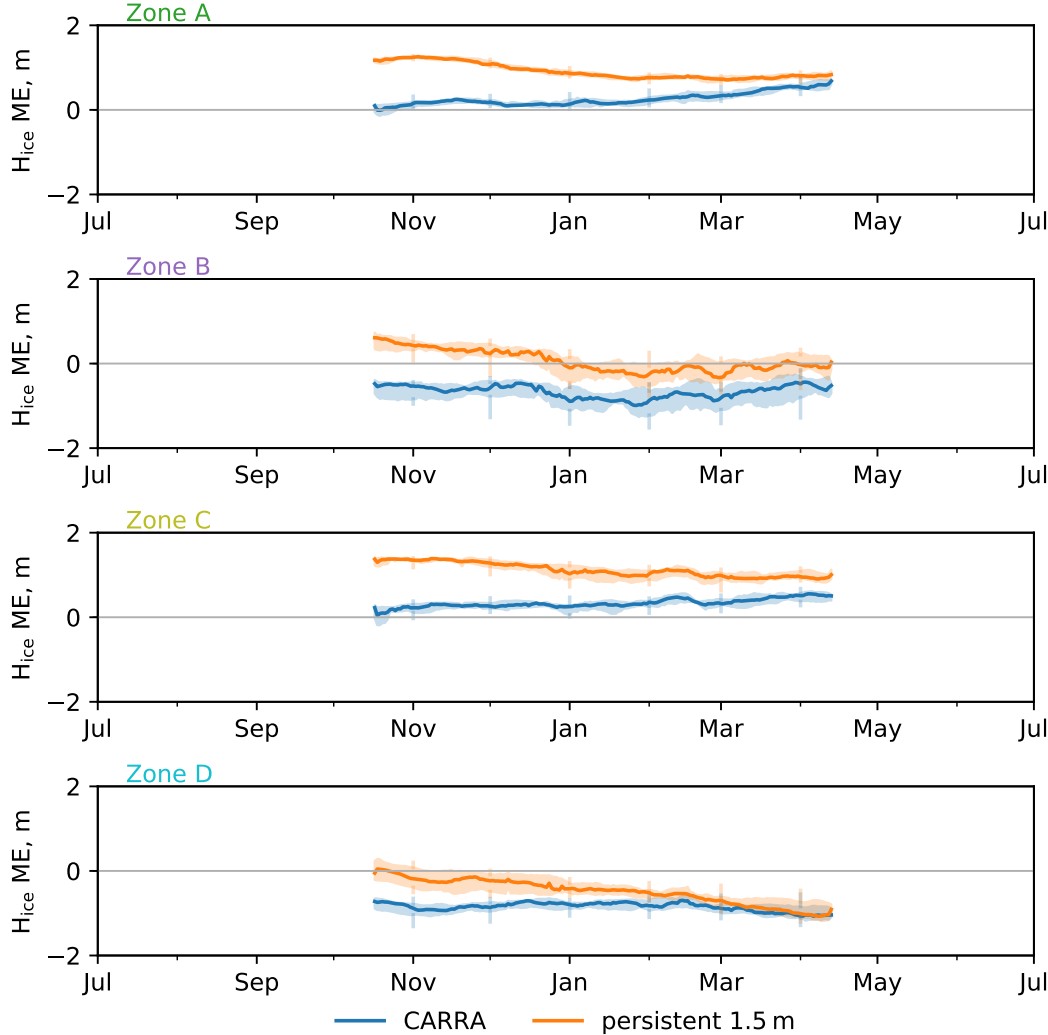

**Figure 8.** Average annual cycle of the mean error (ME) of sea ice thickness in CARRA over the period from 2010 to 2020 for the four areas of interest. On the panels, centre lines show the median ME value, shading bands show the corresponding interquartile ranges, and spikes show the $q \in [0.05; 0.95]$ quantile range. Series of mean error are computed against the combined CryoSat-2 SMOS ice thickness product. Also in the figure, ice thickness errors obtained using constant and uniform ice thickness of 1.5 m as in the ERA5 product are shown. Numerical monthly values of the computed scores can be found in Table S4.

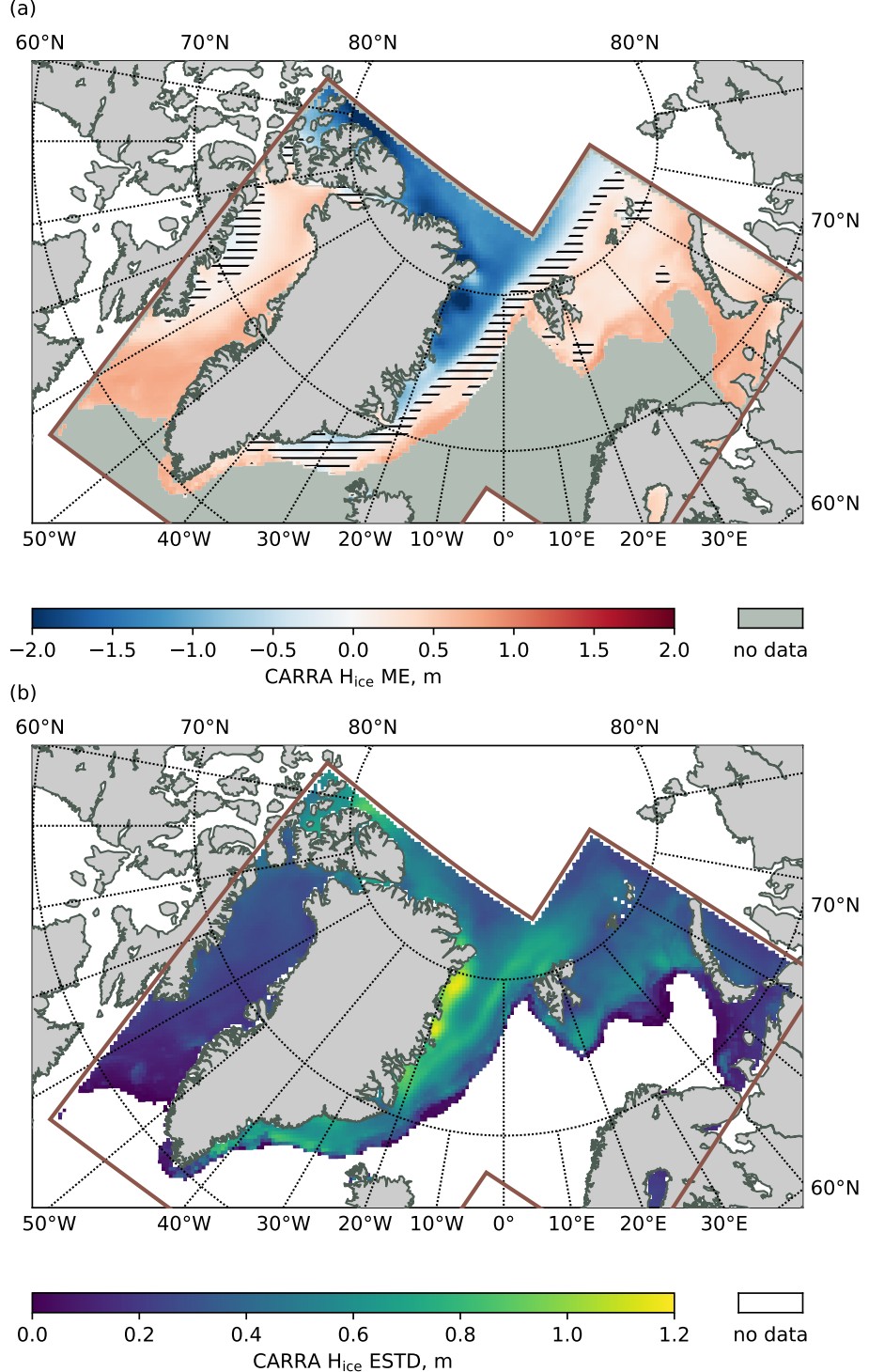

**Figure 9.** Mean error (ME) and standard deviation of errors (ESTD) of sea ice thickness in the CARRA product computed against the combined CryoSat-2 SMOS satellite ice thickness product. (a) Mean error; (b) standard deviation of errors. Also on (a), the areas where ME is below the median uncertainty reported by the product are marked with hatches.

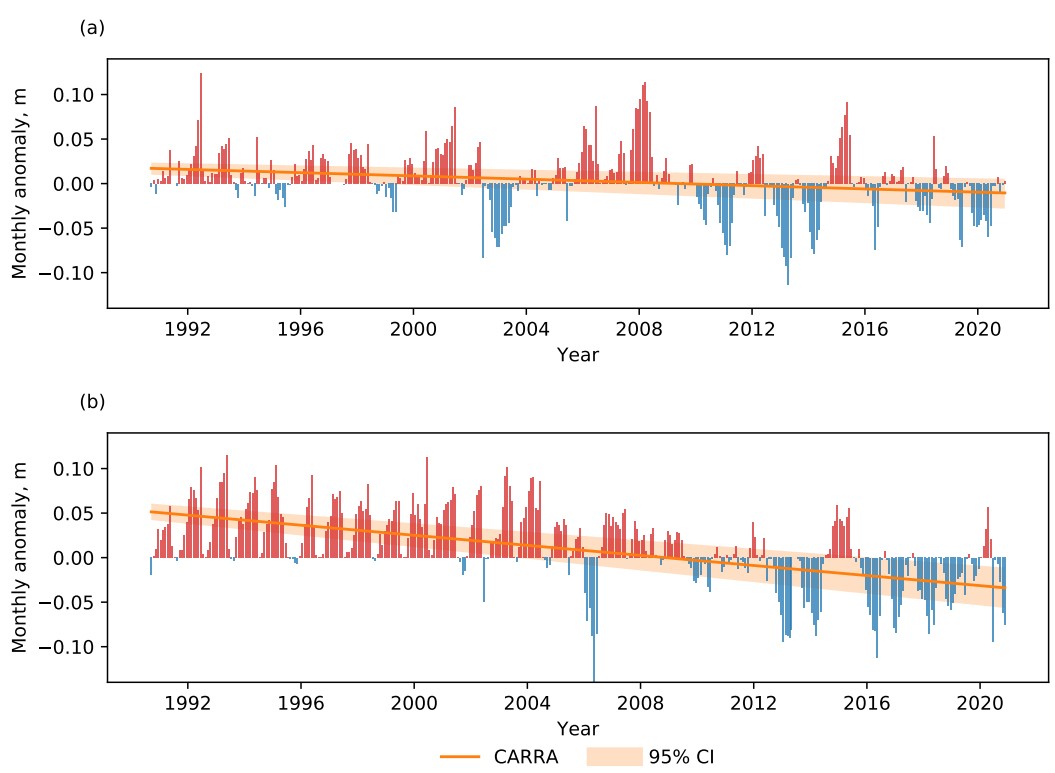

**Figure 10.** Monthly snow depth anomalies in the CARRA product and fitted snow depth anomaly trend. Multiyear monthly means for the time period from 2000 to 2020 are used as a reference when computing anomalies. (a) Western CARRA model domain; (b) eastern CARRA model domain. Also on the panels, the 95% confidence interval of the CARRA anomaly trend is shown.

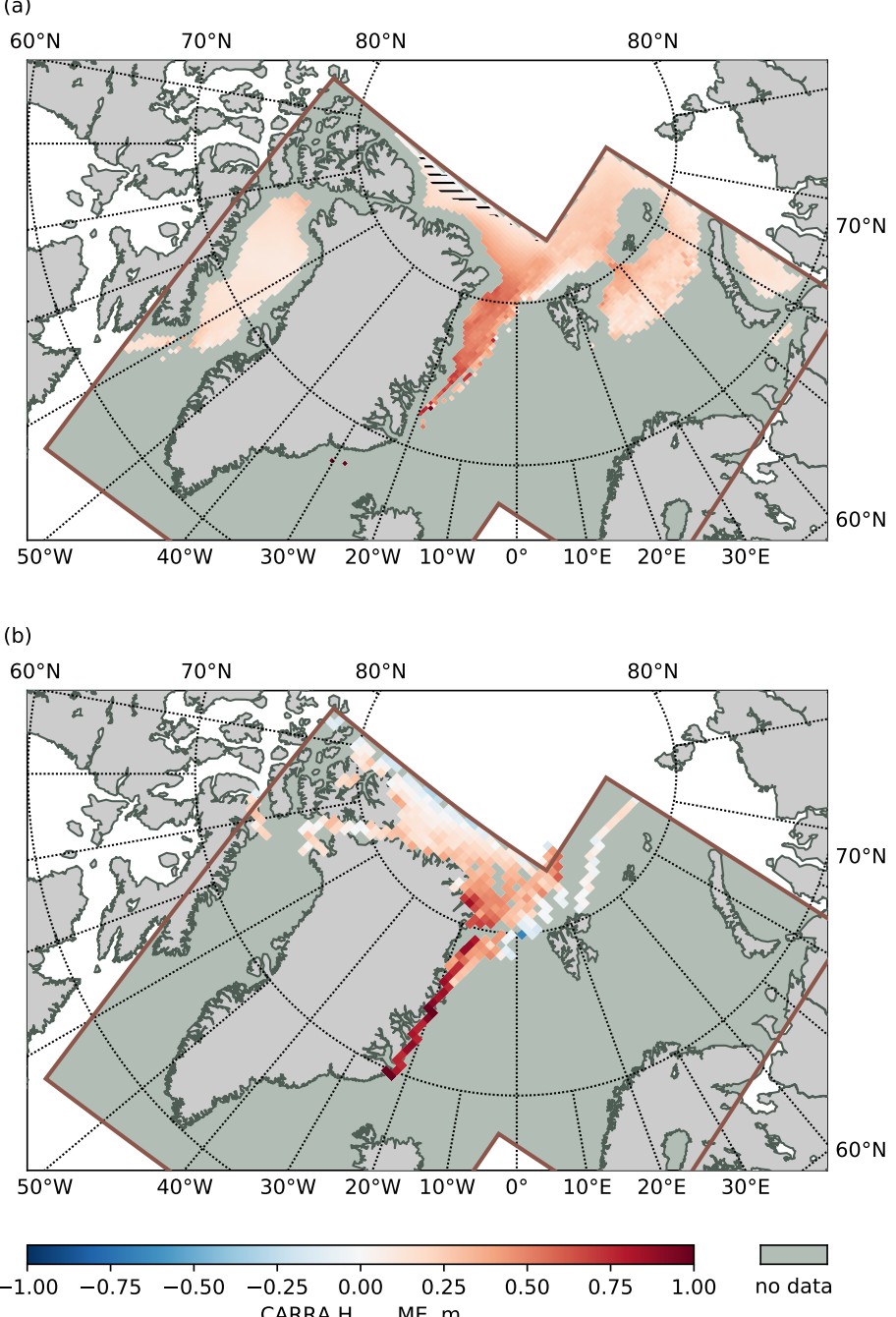

**Figure 11.** Mean error (ME) of snow depth over sea ice in the CARRA product computed against the satellite ice thickness retrieval product and Operation IceBridge flight campaign data. (a) January-March ME computed against the satellite product over the time period from 2003 to 2020; (b) March-April computed against the Operation IceBridge data over the time period from 2009 to 2019 and presented on a 50 km grid. Also on (a), the areas where ME is below the median uncertainty reported by the product are marked with hatches.

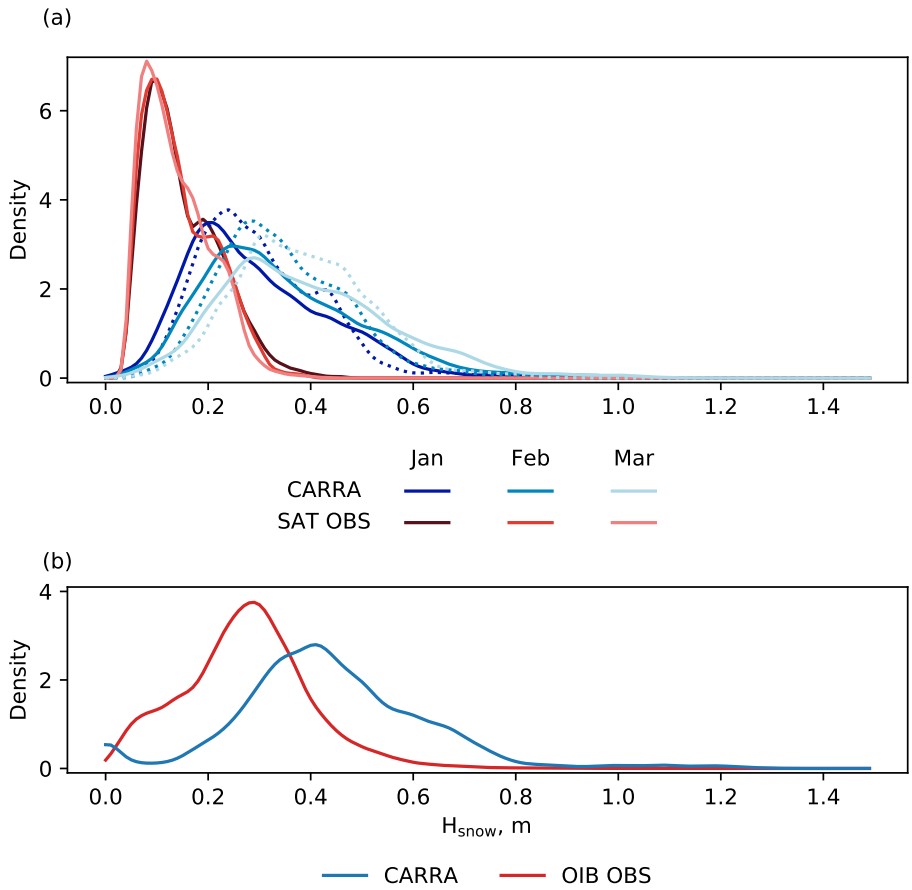

**Figure 12.** Estimated probability density functions (PDF) of snow depth in the CARRA product and in the observational products. (a) monthly PDFs of snow depth in CARRA and in the satellite snow depth retrieval; (b) PDFs of snow depth in CARRA and in the Operation IceBridge flight campaign data. Also on (a), the PDFs of corrected snow depth in CARRA are shown with dotted lines.

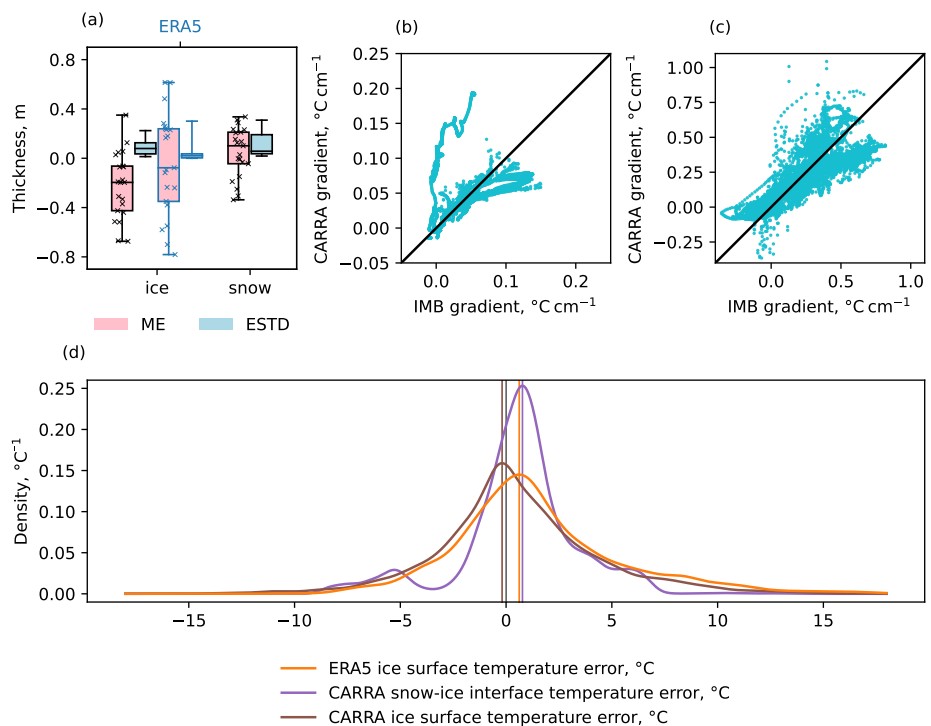

**Figure 13.** Representation of sea ice in the CARRA and ERA5 reanalysis products as compared against buoy observations. (a) Box plots of the per-buoy mean error (ME) and standard deviation of errors (ESTD) of snow depth and ice thickness, with whiskers representing the full range of computed values; (b) temperature gradient within the ice layer computed from buoy data and computed from the CARRA product; (c) temperature gradient within the snow layer computed from buoy data and computed from the CARRA product; (d) estimated probability density functions of ice surface temperature error and snow-ice interface temperature error in CARRA, and ice surface temperature error in ERA5. Also on (d), major modes of the modelling errors in CARRA and ERA5 are marked with vertical bars of corresponding colours.

**Table A1.** Parallel production streams in the CARRA reanalysis project, without taking into account spin-up periods.

| | CARRA West | | CARRA East | |
| --- | --- | --- | --- | --- |
| | Start | End | Start | End |
| BE1 | 1990.09.01 | 1992.08.31 | 1990.09.01 | 1992.08.31 |
| BE2 | 1992.09.01 | 1994.08.31 | 1992.09.01 | 1994.08.31 |
| BE3 | 1994.09.01 | 1997.06.30 | 1994.09.01 | 1997.06.30 |
| S1 | 1997.07.01 | 2002.08.31 | 1997.07.01 | 2006.08.31 |
| S1W | 2002.09.01 | 2006.08.31 | | |
| S2 | 2006.09.01 | 2010.08.31 | 2006.09.01 | 2014.08.31 |
| S2W | 2010.09.01 | 2014.08.31 | | |
| S3 | 2014.09.01 | TU | 2014.09.01 | TU |

All dates are in the Y.M.D format; TU – production stream is continued to produce
near real time updates of the reanalysis product.

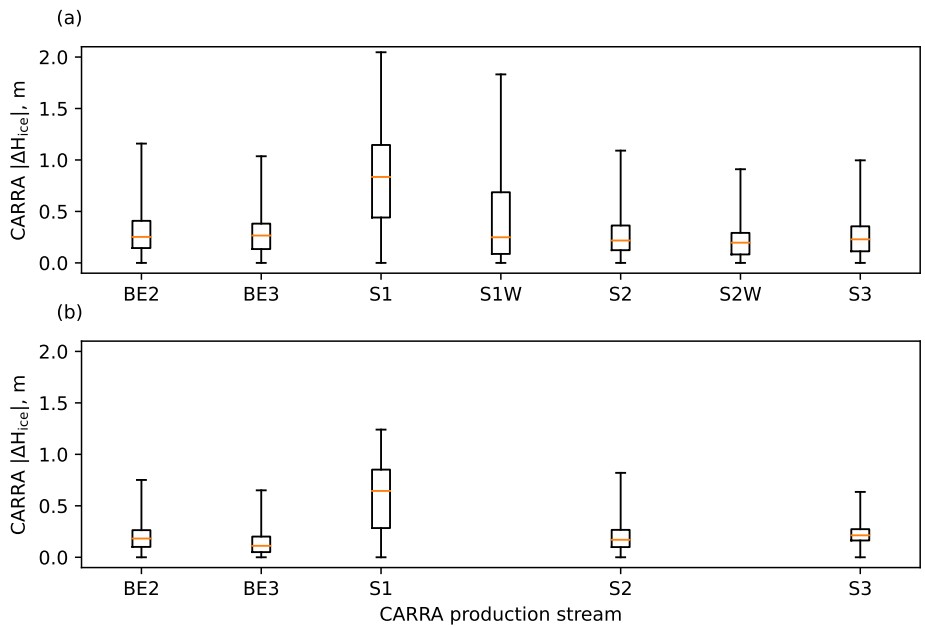

**Figure A1.** Discontinuities in the CARRA ice thickness field at the start of production streams, computed as an absolute value of the ice thickness difference between the last analysis of the ending stream and the first analysis of the starting stream, for grid cells with perennial ice cover. (a) – western CARRA model domain; (b) – eastern CARRA model domain. Whiskers represent the full range of computed values.

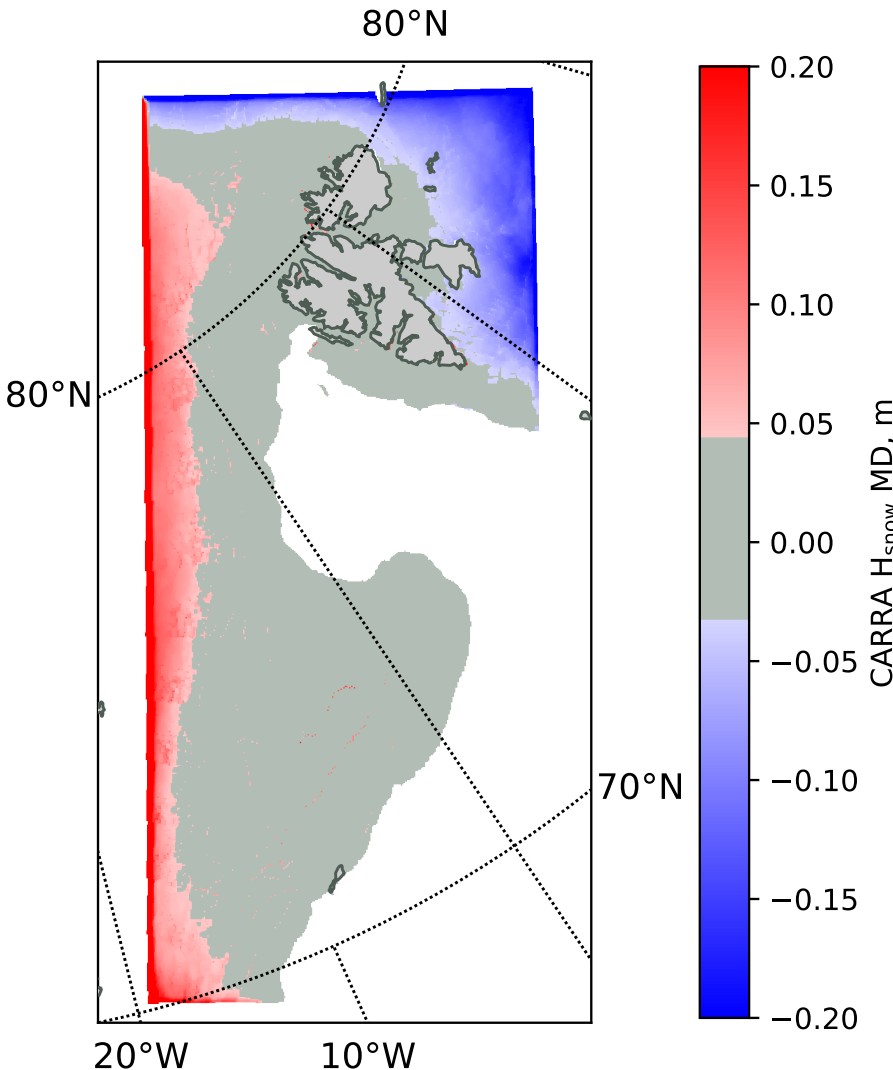

**Figure B1.** Mean difference (MD) of snow depth in two CARRA model domains within the region of geographical overlap, as of 1 April, computed over a period from 1990 to 2021. Positive values mean that snow in the western CARRA model domain is thicker than in the eastern model domain, and vice versa. Grid cells with MD falling within the interquartile range of snow depth differences are masked out.