# Peer review of "Sea ice cover in the Copernicus Arctic Regional Reanalysis"

_The Cryosphere, 2023_

## Author Comment (AC1)

**Response to the referee 1**

Thank you very much for a helpful and constructive review. Please see below for a point-by-point response on your corrections and suggestions (with our replies marked with blue colour and indentation).

The paper is well written and very timely with the recent field work (MOSAiC) that has been undertaken within the Arctic. The paper provides insight into deficiencies with the global reanalysis product for the region, ERA5, and highlights where the regional analysis can provide improved estimates of the surface properties of the sea ice. This is especially useful resource for the modelling community when trying to improve their models of the region. The conclusions provide a good summary of the results and the limitations of the current system which are important for users of this reanalysis. It also gives insight into how future systems can be improved in the future. I think the paper is close to being ready for publication but have a few queries about some details that I think may help to clarify what has been done.

I have very few detailed comments for the authors but a general question I had that made me hesitant to accept as is, was based on understanding the impacts of: the production streams for the reanalysis and the boundary conditions on the results shown.

Two questions come to mind when considering the East and West portions of the reanalysis with multiple production streams and different boundaries. One element is the spin up when you start the production and the impact that has. If I understand correctly, the production streams are not initialised on the same dates for the two different regions, and I wonder whether this provides additional information (or introduces errors in determining the state) in the regions of geographical overlap. The second element is that within the regions of overlap whether particular areas/cells are more constrained by the boundary conditions imposed at the edges of the domain and whether you also see some signature of this comparing similar geographic points in the two analyses. Is this something you looked at all? Can it help provide insight into the performance/constraints of the system? I wondered if when the ice extent was smallest in late summer early autumn and then closer to the edges of the analysis domain is this why the ERA5 and CARRA systems are more similar or if it is a function of the physical processes governing the errors?

> Indeed, due to timeliness constraints the western CARRA domain used more production streams than the eastern one. As a consequence, there are two extra production streams with different initialisation dates. Apart from those two, productions streams in the two CARRA domain were initialised at the same dates. As we mention in our reply to the next comment, across-stream discrepancies are small for all the variables except ice thickness over perennial ice. Therefore, taking into account that ice thickness is a rather peculiar variable in the CARRA product that should be used with great care even outside the stream transition periods, we believe that the two extra production streams in the western CARRA model domain do not significantly affect the representation of sea ice state in the reanalysis.

> As for the impact of boundary effects on the overlapping region and in general, we have not investigated it in detail when preparing the original version of the manuscript. It is true that boundary effects in CARRA are not negligible: firstly, due to the coupling strategy applied in the CARRA system, which does not use hydrometeors from the host model and thus requires some spin up to build up cloud water and ice; secondly, near-surface temperatures over sea ice in ERA5 are generally warmer than in CARRA (in winter time) which also introduces a discrepancy. Assuming that snow depth over sea ice is a variable the most affected by the boundary effects (since it directly depends on model precipitation), and following referee's

[Figure]

Figure 1: Average snow depth difference between western and eastern CARRA model domains as of 1 April, computed throughout the period from 1990 to 2021. Positive values mean that western CARRA domain has thicker snow layer than eastern CARRA domain, and vice versa. Means falling within the interquartile range of snow depth differences are masked out.

suggestion, we assessed them by comparing spring-time snow depths in both CARRA domains over the region of geographical overlap. Figure 1 shows consistent differences in the snow depth and indicates areas of the domains where boundary effects are more pronounced. We believe this information would be of considerable interest to readers and users of the CARRA product and we will provide it in the updated version of the manuscript.

Concerning the late-summer/early-autumn performance of the CARRA system, when differences between the regional reanalysis and ERA5 are smallest, we attribute it to sea ice being in a more constrained state in general rather than to boundary effects. At that time snow cover over sea ice in CARRA has already melted and in absence of the snow layer the sea ice schemes of CARRA and ERA5 become quite similar. Moreover, ice temperature is essentially a bounded variable, and summer ice surface has temperature close to melting point with much less variability compared to cold winter-time ice cover, which further reduces differences between CARRA and ERA5. Similarly, for the ice surface albedo fields, CARRA and ERA5 have less discrepancy over warm summer-time snow-free ice.

In lines 118-119 you talk about the spin up of the snow loading and later in figure 7 you highlight the different production streams used for the E and W components of the reanalysis. Do you have overlapping time periods for the streams that you can compare to determine if the system is spun up? Do you assess the difference where there are overlapping geographical points between the West and East analysis areas given that the production streams restart more frequently in the West - is that also a way of determining if there is any significant impact?. if you have determined that the model has "spun up" within a year please add this to the text. Do the different start dates in the production between the W and E regions means that you need to do more complex averaging e.g. weighted mean for the points which are both in the E and W regions for zones B,C or D when

[Figure]

Figure 2: Evolution of the sea ice state in the CARRA system over a transition period between production streams for a single selected point (82.4°N, 16°E) within the region of geographical overlap. The stream transition occurred for the western CARRA model domain, all values extracted for the eastern model domain come from the same production stream. (a) Ice thickness; (b) snow depth; (c) ice surface temperature.

calculating trends etc. Have you assessed this spin up impact on sea ice thickness as well? Are the production streams always started at the same point in the year (eg. Jan 1st)? Could this have any effect on the climatology estimates shown in figure 8? I wondered whether it might be helpful to expand on this in section 3.1 or in the results section.

The released CARRA data set does not include model output from the spin up parts of production stream and therefore provides no overlapping periods for a selected CARRA model domain. Thus, in the present study we treat the CARRA product as a flat data set and do not try to compensate the differences in productions streams of two CARRA domains by applying weighted averaging over the region of model domain overlap. For most of the sea ice variables discussed in our paper the across-stream difference is rather small since these variables are either 'fast' and considerably influenced by the atmospheric forcing or have a pronounced annual cycle and disappear by the end of summer (like snow cover, or annual ice cover). The only variable which can show a consistently high across-stream discrepancy is the ice thickness, for cases when ice persists for multiple years in a grid cell. Typical performance of the CARRA system across the production streams is shown in Fig. 2. The figure might suggest, that a one

year spin up period used in CARRA production is too short for properly initializing perennial ice cover within the model domain. However, this claim is difficult to validate due to several factors: firstly, sea ice scheme of CARRA is not constrained by observations and may start drifting after years of integration in a production stream, which would introduce across-stream discrepancy even if the spin up period yielded perfect initial state of the ice cover; secondly, the overlapping part of the CARRA model domains, which can be used to assess the spin up effects contains only a minor amount of perennial ice close to the domain boundary, where boundary effects are the most pronounced. Therefore, we believe that in these circumstances, and taking into account limited computational resources available and time constraints of the production phase of the reanalysis project, having a one year spin up period results in adequate initial state of the sea ice thickness field. We assessed potential impacts of across-stream discrepancy on the long-term trends in the ice thickness series (presented in Fig. 7 of the original manuscript) by excluding an extra year of data at the beginning of each production stream, however computed scores were not significantly different from the values reported in the original manuscript. We believe that similarly it would have a minor impact on the annual evolution of errors in modelled ice thickness of CARRA (Fig. 8 in the original manuscript).

Considering your question about the starting date of the CARRA production streams. All but one production streams in CARRA start on 1 September, and one production stream (the first production stream of the initial CARRA reanalysis period, before back extension) starts on 1 July.

We will update the manuscript to document and discuss sea ice spin up and discrepancies across the production streams in CARRA.

Finally, is it possible to show a comparison with the ice surface observations that were made during the MOSAIC drift campaign to show an example of how the reanalysis compares, particularly as there is already a published comparison with the ERA5 and MODIS data by Herrmannsdörfer et al. (2023) for IST.

Indeed, the MOSAiC drift campaign provides very valuable information on the sea ice state in the modern-day Arctic. However, due to the location of the CARRA model domains the drifting station entered them at the late stage of the drift in May 2020, when temperatures rise and melting season starts. In a sense, this period is of somewhat less interest compared to the winter-time one discussed by Herrmannsdörfer et al. (2023) when assessing the model performance, since ice surface temperature in the model (and in observations) becomes close to the melting point of snow/ice and does not wary that much. Therefore, we decided to not include comparisons against MOSAiC in the present study.

Specific comments: Section 2: Lines 110-111: "new ice is always snow free" - I assume that if the ice concentration is updated from a non zero value to a different non zero value (increased/decreased) the snow depth remains the same? or is the snow volume conserved?

Yes, when ice concentration is updated from a non-zero value to another non-zero value, the snow depth within that grid cell remains unchanged. In the updated version of the manuscript we will explicitly mention that 'new ice' there implies the case when ice concentration is changed from zero to a non-zero value and that snow volume is not conserved when ice concentration is being adjusted.

Line 125: IFS-HRES - this acronym is not well explained - perhaps the more relevant information is that it makes use of the ECMWF 4D-Var data assimilation and forecast model

We agree that right now the text assumes that readers are familiar with the IFS-HRES NWP system and lacks details. We will update the manuscript to provide more details as suggested.

Section 3: When you make comparisons with the sea ice thickness and snow depth data have you taken into account the observational error? For the sea ice thickness data the errors can be quite large; the ice thickness retrievals may also be limited by the assumptions made about snow loading.

Indeed, satellite retrievals of ice thickness and especially snow depth can have quite high uncertainty. For this reason, we focused mainly on the qualitative assessment when using these products. However, we have also performed additional checks by comparing the differences between the satellite products and reanalyses against the uncertainty estimates provided with these retrievals. We found that on average the observed differences are above the uncertainty level for most of the area represented in CARRA. We will update our manuscript to explicitly mention that fact.

Section 4 (and figures) Fig 5: Is it possible to plot the ice cover amount somehow with the 4 zones in figure 5? could it be that we are looking at a fairly small area in zones A, B and C near the ice minimum? do you know how much the ERA5 boundary conditions may influence the behaviour of the CARRA as the ice retreats in these regions?

Thank you for a nice idea! You are right, ice-covered area within the four zones of interest can vary considerably throughout the year, and we agree that having this information could be beneficial. We will try to incorporate it in some form into the updated version of the manuscript.

Considering your second question, we believe that ERA5 boundaries do not significantly affect sea ice state in CARRA when ice retreats since it happens during the summer melt season. At that time (when ice have retreated considerably enough to remain only in the areas close to the model domain boundary) sea ice and snow cover on top are well into the melting regime with temperatures close to 0°C in both CARRA and ERA5 which reduces the difference between two products and thus potential influence of the boundary conditions.

Fig 13: What are the whiskers representing in the box and whisker plots (standard deviation, max/min values?)

For this figure in the original version of the manuscript we used a common form of a box plot where whiskers represent the distance to the farthest data point still located within the $1.5 \times \mathrm{IQR}$ distance (where $\mathrm{IQR}$ stands for "interquartile range") counted from the box's top/bottom. In this specific case the applied procedure yielded no outliers for all the box plots except the one representing the standard deviation of ice thickness errors in ERA5, which means that in these box plots with no outliers whiskers show the min/max range of the processed sets. To make the figure less cumbersome we will update the box plots to always represent the min/max range of input data and explicitly mention that in the figure caption.

Lines 565-566: you comment on the accumulation of the snow - do you have a sense of where the deficiency is in the model - is it the lack of the advective processes in the model?

We believe that in this specific case, the observed underestimation of snow depth in CARRA is more likely caused by boundary effects rather than by the lack of the advective processes in the sea ice parameterisation scheme. In the beginning of August 2012, when the IMB reports the onset of new snow layer accumulation, the buoy was located relatively close to the edge of the model domain where boundary effects (such as precipitation spin up) are not negligible. This can be illustrated by Fig. 3 which shows a considerably later start of snow accumulation in CARRA compared to ERA5. On the other hand, representing sea ice dynamics in CARRA would have had only a limited effect on the evolution of snow layer in this case as suggested by relatively small difference between snowfall accumulated along the IMB drift trajectory and

[Figure]

Figure 3: Accumulated snowfall in CARRA and ERA5 over a period from 1 August 2012 to 24 October 2012, extracted along the IMB drift trajectory. Solid lines represent total snowfall amount accumulated in a grid cell between 1 August and the date when IMB entered that grid cell. Dashed lines show snowfall accumulated along the IMB drift trajectory. Also in the figure, snow depth reported by IMB is outlined.

in stationary model grid cells (see Fig. 3). However, individual IMBs show a lot of variability and provide highly local observations therefore in our manuscript we decided to focus on the summary statistics across a set of buoys and to not provide detailed case studies on individual IMBs. We will update the manuscript to properly mention potential effects induced by the model boundaries in CARRA.

**References**

Herrmannsdörfer, L., Müller, M., Shupe, M. D., and Rostosky, P.: Surface temperature comparison of the Arctic winter MOSAiC observations, ERA5 reanalysis, and MODIS satellite retrieval, Elementa: Science of the Anthropocene, 11, 00 085, https://doi.org/10.1525/elementa.2022.00085, 2023.

---

## Author Comment (AC2)

**Response to the referee 2**

Thank you very much for a helpful and constructive review. Please see below for a point-by-point response on your corrections and suggestions (with our replies marked with blue colour and indentation).

This manuscript assesses four variables in the CARRA data: ice surface temperature, ice albedo, ice thickness and snow depth on sea ice, against satellite data, in-situ observation, and the ERA5 data. It is found that both improvement and deterioration, mainly regional dependent, exist in the performance of the target variables in the CARRA data in comparison to the ERA5 data.

In general, the manuscript is significant and valuable for the users of the CARRA data, in this sense, I recommend major revision for this version of the manuscript. Substantial revision is needed before the manuscript can be accepted for publication: 1) the manuscript lacks possible mechanism analysis linking the observed improvements of the four variables to the differences in sea ice scheme between the CARRA and ERA5 systems, this part could be arranged in the last section. 2) The manuscript needs English-editing, some sentences have illogical structures, which make the readers hard to follow.

> Similar concerns regarding limited analysis of the relations between differences in sea ice parameterisation schemes of CARRA and ERA5 and observed performance of the two reanalysis systems were raised by the editor. We agree that the submitted manuscript is somewhat lacking in that aspect, mainly because for two out of the four variables of interest (namely for ice thickness and snow depth) ERA5 does not represent the governing processes at all, and for the other two either the reasons for observed improvement in CARRA are covered by earlier studies (for ice surface temperature) or not clear improvement is found (for ice surface albedo). Nevertheless, we will update the manuscript to provide more discussion on how the observed improvement/degradation found in CARRA compared to ERA5 is connected to the differences in the representation of the sea ice cover in the two reanalysis systems.

> Concerning the language and sentence structure of the manuscript, we will revise the text to improve its clarity and readability.

Minor comments:

L14: Over the whole MS, the authors use some statements like "added value", however, such statements are unclear. It is better to directly use: the performance of CARRA on XXX is better/worse than the ERA5.

> In our paper we use the term 'added value' to highlight and acknowledge that, in fact, CARRA, as a regional reanalysis, is forced by the ERA5 data. However, this term is used only once in a sense 'CARRA is better than ERA5' while in other cases we refer to (or imply) '*potential* added value' and use the term in the introductory sections when performance of the two reanalyses is yet to be assessed. When performing actual comparison we already use direct terms and state whether CARRA is better/worse than ERA5.

L42: Can you explain why operational NWP systems are tuned to perform best in midlatitudes? Why not in Tropics?

> In the original manuscript when we state that operational NWP systems are tuned to perform best in mid-latitude regions we meant the cases when a model domain includes both mid-latitude and polar regions. Obviously, that statement does not apply for an operational weather forecasting system used over a tropical model domain where it makes little sense to tune that

NWP system based on its mid-latitude or polar performance. We will update the text to convey our point in a more clear way and to avoid confusing generalisations.

L50: Can you specify what is "simplified one-dimensional parameterisation schemes"? This statement is so blurry.

We used 'simplified one-dimensional parameterisation schemes' as an umbrella term to distinguish simple schemes (which could be as basic as prescribed ice surface temperature, or thermal balance of a thin ice layer) from more advanced thermodynamic and dynamic-thermodynamic sea ice models. In the following paragraph of the main text in the original version of the manuscript we provided a number of examples of such schemes. In the updated version of the manuscript we will additionally refer to Hines et al. (2015), Køltzow et al. (2019) and Solomon et al. (2023), should the readers be interested in simple sea ice parameterisation schemes applied in operational non-coupled NWP systems.

L65: "Additionally" is not needed.

We believe that sentence is more readable as it as is now, therefore we will keep it in the revised manuscript.

L85, 124, 238, 620: "at the moment of writing this manuscript" is not needed.

We believe that these remarks are well-justified and necessary to not confuse readers if some of the discussed products would be extended or cease operational production after this study is published, so we will keep them in the revised version of the manuscript.

L85: with a time interval of 3 hours

We agree that 'every 3 hours' in the original manuscript looks somewhat confusing. We will reword that sentence.

L85-87: "the CARRA data set includes the output from model integration...lead times over 6 hours". Does this mean the CARRA also include forecasting data? If so, this statement could be changed to "the CARRA data set also includes model forecasting data......."

Yes, the CARRA data set provides both objective analysis and forecasting fields. To make it more clear we will update the manuscript as suggested.

L101-102: "Sea ice albedo ....surface albedo". As you mentioned direct albedo, diffuse albedo, it is better to declare their definition, otherwise you can obsolete this sentence.

To us the terms 'direct albedo' and 'diffuse albedo' are pretty self-explanatory and do not require additional introduction. However, we agree that it might be somewhat confusing to see these terms for a reader more used to 'black-sky' and 'white-sky' albedo. We will update the manuscript to make it more clear, and refer to Lucht et al. (2000) in case readers are interested in additional details.

L114: Can you explain what is "fallback data set"?

In the manuscript we refer to OSISAF OSI-450 as a 'fallback data set' meaning that it is used for dates when ESA CCI SICCI data are not available. We will update the text to make it more clear.

L140: Please illustrate how to calculate verification score. Or give an reference.

Our intention was to simply state that ERA5 is used as a baseline when assessing performance of CARRA (where applicable) without referring to any specific metric. We will reword that sentence to make it more clear. As for the actual verification scores discussed in the our paper, we use pretty standard statistics such as mean error or standard deviation of error which, we believe, do not require additional explanations.

L143: for example

Will be corrected.

L147-148: "in the present study..... a denser model grid"....hard to follow. Please reorganize.

Indeed, that sentence is rather cumbersome and fits poorly within the rest of the paragraph. We will remove this sentence in the updated manuscript and extend the paragraph to be more clear and easy to follow.

L150: but the characteristics

We think that in this context 'and' would be a better replacement for 'although' used in the original version of the paper. We will update the manuscript accordingly.

L161: Where is White Sea? Not marked in Fig. 1.

Thank you for spotting a missing label, we will update the figure to include it.

L168: "of the the sea ice"......delete one of "the"

We will update the manuscript accordingly.

L169: those derived

Will be corrected as suggested.

L190: "MODIS retrieves .... in situ observations". Any reference ???

To support our statement that MODIS ice surface temperature retrievals used in the present study tend to show a cold bias when compared to in situ observations, we will refer to studies by Hall et al. (2004), Herrmannsdörfer et al. (2023) and Li et al. (2020) in the updated version of the manuscript.

L218: Why the data between 2016 and 2019 is not used?

In the present study most of the analysis is focused on assessing the performance of the CARRA system in representing the modern-day Arctic. Therefore a representative 15 year period (from 2000 to 2015) was chosen, despite more data being available for both CARRA and CLARA-A2. Also, the 2016-2019 part of CLARA-A2 has been done as an extension. In this extension aerosol quantities are taken as climatology values, meaning the intercalibration of the radiances is not quite as robust as in the non-extension part up to 2015. The difference between the main data set and the extension is not very big, but considering that characteristic biases of CARRA were already apparent from the initial comparisons, we decided to exclude the 2015 onwards extension.

L362: January-February and middle August-middle October in zone A, and middle August-September in zone B

Thank you for providing these corrections, we agree with all of them except the first one. In February over zone A CARRA shows slightly lower (in absolute value) ice surface temperature error compared to ERA5 (as can be seen from Table S3). We will update the manuscript accordingly.

L364: December to March

To make that sentence less ambiguous, we will change the text from '*from December to April*' to '*from December to the end of March*'.

L369: December to March

Similarly to the previous comment, we will explicitly state in the updated manuscript that the period spans from December to the end of March.

L411: no supporting evidence for the statement "which is attributed ... central Arctic"

We think that our word choice might have confused the referee, as we used 'central Arctic' to refer to the northernmost part of the CARRA model domains. We agree that, as it is written now, that sentence could be interpreted in a way that snow accumulation and temperature drop in the central Arctic *outside* the CARRA model domains influences the surface albedo evolution *within* the CARRA area. Indeed, such a statement would require strong supporting evidence. However, in our manuscript we do not speculate about potential impacts of the outside regions on the performance of the reanalysis system but simply explain the observed feature in the distribution of sea ice albedo errors in August based on our knowledge of the technical implementation of the sea ice parameterisation scheme applied in CARRA with two different albedo formulations for snow-free and snow-covered sea ice grid cells. In the beginning of August sea ice in the model is actively melting and and dark snow-free ice surface is exposed (there are also grid cells which still retain some amount of snow, but they also have relatively low albedo of melting snow even though not as low as for melting ice). However, by mid-August new snow starts covering the ice resulting in increased surface albedo of snow-covered grid cells. Typically, by the end of August a considerable fraction of sea ice within CARRA model domains is covered by new snow, although the snow-free part is still not negligible. This results in a bimodal distribution of August sea ice albedo error in CARRA with one mode corresponding to bright fresh snow and another one — to grid cells with dark ice and old melting snow. This situation is illustrated in Fig. 1 showing the median snow extent over sea ice at the end of August. Therefore, to support our statement we will update the albedo error maps figure in the revised version of the manuscript to show the August snow extent. Additionally, we will clarify in the text that 'central Arctic' refers to the northernmost part of CARRA model domains to make the text more clear and avoid further confusion.

[Figure]

August

Sea ice albedo error, %

no data

Figure 1: Mean error of the modelled surface albedo in August over sea-ice covered regions of CARRA, computed against the CLARA-A2 SAL product over the time period from 2000 to 2015. Median snow extent at the end of the month is outlined by a contour.

L440: delete ", for example"

We will update the manuscript accordingly.

L499: "first years"???

We meant the first half of the snow depth anomaly series there. We will reword that sentence to make it more clear.

**References**

Hall, D., Key, J., Casey, K., Riggs, G., and Cavalieri, D.: Sea ice surface temperature product from MODIS, IEEE Transactions on Geoscience and Remote Sensing, 42, 1076–1087, https://doi.org/10.1109/TGRS.2004.825587, 2004.

Herrmannsdörfer, L., Müller, M., Shupe, M. D., and Rostosky, P.: Surface temperature comparison of the Arctic winter MOSAiC observations, ERA5 reanalysis, and MODIS satellite retrieval, Elementa: Science of the Anthropocene, 11, 00 085, https://doi.org/10.1525/elementa.2022.00085, 2023.

Hines, K. M., Bromwich, D. H., Bai, L., Bitz, C. M., Powers, J. G., and Manning, K. W.: Sea Ice Enhancements to Polar WRF, Monthly Weather Review, 143, 2363–2385, https://doi.org/10.1175/MWR-D-14-00344.1, 2015.

Køltzow, M., Casati, B., Bazile, E., Haiden, T., and Valkonen, T.: An NWP Model Intercomparison of Surface Weather Parameters in the European Arctic during the Year of Polar Prediction Special Observing Period Northern Hemisphere 1, Weather and Forecasting, 34, 959–983, https://doi.org/10.1175/WAF-D-19-0003.1, 2019.

Li, N., Li, B., Lei, R., and Li, Q.: Comparison of summer Arctic sea ice surface temperatures from in situ and MODIS measurements, Acta Oceanologica Sinica, 39, 18–24, https://doi.org/10.1007/s13131-020-1644-7, 2020.

Lucht, W., Schaaf, C., and Strahler, A.: An algorithm for the retrieval of albedo from space using semiempirical BRDF models, IEEE Transactions on Geoscience and Remote Sensing, 38, 977–998, https://doi.org/10.1109/36.841980, 2000.

Solomon, A., Shupe, M. D., Svensson, G., Barton, N. P., Batrak, Y., Bazile, E., Day, J. J., Doyle, J. D., Frank, H. P., Keeley, S., Remes, T., and Tolstykh, M.: The winter central Arctic surface energy budget: A model evaluation using observations from the MOSAiC campaign, Elementa: Science of the Anthropocene, 11, 00 104, https://doi.org/10.1525/elementa.2022.00104, 2023.

---

## Author Comment (AC3)

**Response to the editor comments**

Thank you very much for helpful and constructive comments. Please see below for a point-by-point response on your comments (with our replies marked with blue colour and indentation).

Both reviewers pointed out potentials of this study but also provide constructive comments which I believe are valuable for authors to improve the manuscript. In addition to comments from two reviewers, I have a few comments for authors to consider during the revision:

I got impression authors have done a lot work in this study and presentation of results were comprehensive. However, the reasons of "why" seems a bit weak. The authors claims the biggest improvement was the surface temperature calculation: "The strongest improvement was observed for winter months over the Central Arctic, and the Greenland and Barents seas where a 4.91°C median ice surface temperature error of ERA5 is reduced to 1.88°C in CARRA on average." In winter months, the stable boundary layer (SBL) may dominate the surface energy balance. CARRA seems tackle SBL better than ERA5, yet the stable boundary layer was not mentioned in the manuscript at all. I would like to see authors give some discussion on it.

> A similar comment was given by the second referee and we agree that the original manuscript provides only limited discussion on the mechanisms and links between the observed verification results and differences in representing sea ice in CARRA and ERA5. In fact, for ice surface temperature CARRA's implementation relies on the findings of an earlier study (Batrak and Müller, 2019) that assessed the impacts of utilising snow cover in the sea ice parameterisation scheme of HARMONIE-AROME, therefore we find it somewhat excessive to repeat that study once again. For other variables, namely ice thickness and snow depth, ERA5 simply does not represent their evolution and any improvement/degradation found in CARRA is purely because of the additional prognostic formulations included in the regional reanalysis systems. Nevertheless, we agree that our paper can benefit from extending the discussions and we will update the manuscript accordingly.

> Concerning your comment about the representation of winter-time SBLs in CARRA compared to ERA5. Indeed, a snow layer with low thermal conductivity allows for more effective radiative cooling of the ice surface and stronger inversion. This should aid in a better representation of SBLs, however in absence of reliable observational data we can not confirm that with certainty. In general, applying a more detailed sea ice scheme impacts many atmospheric variables from two metre temperature and turbulent fluxes directly over sea ice to cloud cover and precipitation tens and hundreds kilometres away from the ice edge during cold air outbreaks. Assessing all the impacts induced by the ice cover in one study is simply impossible. Therefore, we decided to focus only on the sea ice variables and leave investigations of atmospheric parameters to future studies.

Several places in the manuscript, I would like to see some discussion on "why" e.g.
L630 "The sea ice cover in CARRA adequately represents general multiyear trends towards thinner and warmer ice cover, connected to the ongoing climate change in the Arctic. Comparisons against the satellite-based and in situ sea ice observations show generally improved representation of sea ice in CARRA (using ERA5 as a baseline), although this improvement is not universal". Why?

> We are somewhat puzzled by this comment. This sentence simply says that CARRA is not always better than ERA5 and we do not see how we can further extend on this.

"The main difference between the sea ice schemes in ERA5 and CARRA is the presence of an explicitly resolved snow layer, which allows for much lower ice surface temperature in the CARRA

system, therefore reducing the warm ice surface temperature bias found in ERA5" see my comment on SBL above. Can we say that the presence of a snow layer can better tackle the SBL?

> Well, indeed, having a snow-covered ice surface results in strong radiative cooling in winter-time clear-sky conditions thus allowing for much stronger inversion compared to snow-free ice surface. However, it is difficult to draw conclusions on how much SBLs are improved in CARRA compared to ERA5 without any reference data. Therefore, a dedicated study would be needed to thoroughly investigate the representation of the atmospheric boundary layer in CARRA. We will update the manuscript to mention potential positive impacts of lower ice surface temperature on representation of SBL in CARRA.

" However, for the area covering Baffin Bay and the Davis Strait the verification scores suggest that a warm winter-time bias of ERA5 is replaced with a cold bias in CARRA." Why?

> We believe that for Baffin Bay and the Davis Strait snow depth and ice thickness in CARRA are overestimated which leads to a cold bias in the ice surface temperature. We will update the manuscript to explicitly state that.

"No improvement over ERA5 was found in the ice surface albedo with spring-time errors in CARRA being up to 8% higher on average than those in ERA5 when computed against the CLARA-A2 satellite retrieval product." Why?

> This result is a simple consequence of different albedo parameterisation schemes applied in CARRA and ERA5. We will update the manuscript to note that.

L95: "Surface albedo of the sea-ice covered grid cells in CARRA is computed by applying simple parameterisation schemes. For snow-free ice cover, a temperature-dependent broadband albedo scheme is applied (defined as HIRHAM in Liu et al., 2007), and when ice is covered by snow an adapted version of the broadband snow albedo scheme by Douville et al. (1995) is used. When computing albedo of cold dry snow covering sea ice in the CARRA system, the albedo scheme of Douville et al. (1995) is modified to increase the value of the lowest possible albedo in the dry albedo degradation term from the original 0.5 to 0.75. Sea ice albedo schemes applied in CARRA do not distinguish between direct and diffuse components of surface albedo. The HARMONIE-AROME NWP system does not produce grid-cell averaged albedo as an output variable, therefore in the CARRA product the surface albedo field is computed from the hourly accumulated downwelling and upwelling shortwave radiation fluxes and available only from the model integration output."

Are you saying for each CARRA grid, in ice covered area, the surface albedo was calculated by simple parameterization schemes. For ice free area, the surface albedo was calculated by the ratio of upwelling /downwelling shortwave radiative flux, right? Then what is the final product of surface albedo for each CARRA grid cell? Is this (albedo in CARRA cell) better to be called "CARRA surface albedo field"? Can this ice surface albedo and ice-free surface albedo played any role on no improvement of CARRA surface albedo over ERA5 product?

> No, we meant something different there. Surface albedo in the HARMONIE-AROME NWP system is always computed by parameterisation schemes for all surface types found within a grid cell, be it ice-covered or ice-free sea, or land. Then, a grid-cell average albedo is also computed and reported to the atmospheric components of the model. However, even though HARMONIE-AROME technically has a grid-cell average albedo as one of its internal parameters, the NWP system does not write it to the output files. So, to have this field available in the CARRA product it had to be reconstructed from the downwelling and upwelling shortwave radiation output fields produced by HARMONIE-AROME. This is a purely post-processing procedure and it has no effect on the performance of the model itself. We will update the text to make it more clear.

**References**

Batrak, Y. and Müller, M.: On the warm bias in atmospheric reanalyses induced by the missing snow over Arctic sea-ice, Nature Communications, 10, 4170, https://doi.org/10.1038/s41467-019-11975-3, 2019.